# A live cell reporter of exosome secretion and uptake reveals pathfinding behavior of migrating cells

Bong Hwan Sung [1], Ariana von Lersner[2], Jorge Guerrero[3], Evan S. Krystofiak[4], David Inman[3], Roxanne Pelletier[1], Andries Zijlstra [2,5], Suzanne M. Ponik[3] & Alissa M. Weaver[1,2,5 ✉]

Small extracellular vesicles called exosomes affect multiple autocrine and paracrine cellular phenotypes. Understanding the function of exosomes requires a variety of tools, including live imaging. Our previous live-cell reporter, pHluorin-CD63, allows dynamic subcellular monitoring of exosome secretion in migrating and spreading cells. However, dim fluorescence and the inability to make stably-expressing cell lines limit its use. We incorporated a stabilizing mutation in the pHluorin moiety, M153R, which now exhibits higher, stable expression in cells and superior monitoring of exosome secretion. Using this improved construct, we visualize secreted exosomes in 3D culture and in vivo and identify a role for exosomes in promoting leader–follower behavior in 2D and 3D migration. Incorporating an additional non-pH-sensitive red fluorescent tag allows visualization of the exosome lifecycle, including multi-vesicular body (MVB) trafficking, MVB fusion, exosome uptake and endosome acidification. This reporter will be a useful tool for understanding both autocrine and paracrine roles of exosomes.

---

[1] Department of Cell and Developmental Biology, Vanderbilt University School of Medicine, Nashville, TN, USA. [2] Program in Cancer Biology, Vanderbilt University School of Medicine, Nashville, TN, USA. [3] Department of Cell and Regenerative Biology, School of Medicine and Public Health, University of Wisconsin-Madison, Madison, WI, USA. [4] Vanderbilt University Cell Imaging Shared Resource, Nashville, TN, USA. [5] Department of Pathology, Microbiology and Immunology, Vanderbilt University Medical Center, Nashville, TN, USA. ✉email: alissa.weaver@vanderbilt.edu

Extracellular vesicles (EVs) are nano-sized vesicles released from cells with potent autocrine and paracrine biological activities[1]. Although EVs were first considered to be cell debris with little biological relevance, EVs are now understood to constitute a fundamental mode of cell—cell communication that mediates delivery of specific protein, nucleic acid and lipid cargoes to recipient cells in diverse contexts[1,2]. Indeed, EVs are involved in the pathogenesis of diverse diseases, including infections[3], neurodegenerative disorders[4], cardiovascular disease[5], and cancer[6,7].

Extracellular vesicles can be classified by their size, biogenesis mechanism, cargoes, and density, e.g. small EVs, including exosomes, and larger EVs such as shed microvesicles (MVs) and large oncosomes[8]. Exosomes are a type of small EV with diameter of 50—150 nm and have been the most studied with respect to their involvement in EV function[1,2]. Exosomes are formed as intraluminal vesicles (ILVs) in late endosomal organelles called multivesicular bodies (MVBs) and secreted after fusion of MVBs with the plasma membrane. Several exosome biogenesis processes have been proposed, including capture of ubiquitinated cargoes by the endosomal sorting complex required for transport (ESCRT) machinery[9]. Syndecan-syntenin complexes also regulate exosome biogenesis via the ESCRT accessory protein Alix[10]. An ESCRT-independent biogenesis mechanism involving membrane curvature induced by ceramide generation by neutral sphingomyelinase 2 (nSMase2) has also been described[11]. Tetraspanin proteins such as CD9, CD63, and CD81 are frequently used markers of exosomes and other small EVs and may be involved in ILV cargo selection and/or biogenesis[12].

CD63 is a member of the tetraspanin superfamily, is highly enriched on ILVs in late endosomal MVBs, and is a broadly used classic exosomal marker. Knockout expression of CD63 by CRISPR/Cas9 impairs secretion of small EVs but not large EVs, suggesting that it contributes to exosome biogenesis[13]. CD63 has been used to label and track exosomes and MVBs in many studies[14–18]. However, most previous studies used pH-insensitive fluorescent proteins such as GFP or RFP, leading to extremely bright fluorescence of internal endosomes. This bright internal fluorescence limits the ability to observe fusion events of MVBs with the plasma membrane due to a poor signal-to-noise ratio. To solve this problem, we adapted an approach from the synaptic vesicle field, that leverages the properties of a pH-sensitive GFP derivative, pHluorin, to observe dynamic vesicle fusion events[19]. pHluorin is virtually nonfluorescent under acidic conditions but fluoresces at neutral pH[19], making it an ideal reporter to observe the fusion of acidic late endosomal MVBs with the plasma membrane. As the outside of the cell is at neutral pH, the acid to neutral switch occurs at the fusion step. We first developed a pHluorin-tagged CD63 reporter to track exosome secretion and used it to demonstrate that MVB fusion precedes adhesion formation in spreading cells by 1—2 min[20]. We also observed that pHluorin-CD63-positive adhesive trails were left behind migrating cells[20]. Subsequently, a similar reporter, with the pHluorin group placed 7 amino acids away from ours in the first extracellular loop of CD63, was used to study regulation of exosome secretion by G-protein-coupled receptor signaling[21].

pHluorin-CD63 is a powerful tool to track exosome secretion and MVB fusion with the plasma membrane. However, while this construct is useful in 2-dimensional (2D) tissue culture conditions with high-resolution imaging, the pHluorin moiety is subject to degradation in cells[22], leading to low levels of expression and fast photobleaching. Thus, we have found it to be difficult to stably express in cells, which limits its use for live imaging to select conditions.

In this study, we improved the stability and brightness of supereclliptic pHluorin-CD63 by incorporating a single amino acid mutation, M153R, previously identified to stabilize ratiometric pHluorin in bacterial fusions[22]. We demonstrate that the mutated pHluorin-CD63, pHluo_M153R-CD63, can now be expressed as a stable construct in cells and is a bright reporter for exosome secretion. Using this construct, we are able to easily visualize exosome puncta and observe pathfinding behavior of migrating cells along extracellularly deposited exosome trails in both 2D and 3D. By tagging an additional pH-insensitive red fluorescent protein to pHluo_M153R-CD63, we are further able to track multiple aspects of the exosome lifecycle, including MVB movements within cells before fusion, endocytosis of extracellular exosome deposits, and acidification of exosome-containing endocytic compartments.

## Results

**pHluo_M153R-CD63 is a bright, stable live imaging reporter**. To improve on our previous reporter, we tested whether a mutation previously shown to stabilize ratiometric pHluorin in bacterial fusion proteins, M153R[22], would also stabilize our supereclliptic pHluorin-CD63 construct. The pHluorin-CD63 construct from our previous study[20] was mutated on Methionine 153 to Arginine by site-directed mutagenesis (pHluo_M153R-CD63, Fig. 1a). As shown in Fig. 1b, CD63 is a tetraspanin protein with two extracellular loops. The pHluorin is inserted into the first small extracellular loop at position 43. Upon fusion of MVB with the plasma membrane, the pHluorin moiety is exposed to neutral pH and becomes fluorescent, which enables dynamic monitoring of exosome secretion (Fig. 1b). After site-directed mutagenesis, pHluo_M153R-CD63 was cloned into a lentiviral vector and stably expressed in HT1080 fibrosarcoma cells. To test whether pHluo_M153R-CD63 labels small EVs, conditioned media of pHluo_M153R-CD63-expressing HT1080 cells were collected and serially centrifuged. Large EVs, typically consisting of shed MVs, were pelleted through a $10,000 \times g$ spin for 30 min and small EVs, typically containing exosomes, were pelleted by centrifugation at $100,000 \times g$ overnight. Nanoparticle tracking analysis (NTA) of small EVs showed the expected size distribution for exosomes with a peak diameter of 105 nm whereas large EVs had peak diameters of 195 and 405 nm (Fig. 1c). Consistent with the previously reported role of pHluorin-CD63 as a reporter of MVB fusion and exosome secretion, immunoblotting of cell lysates and purified EVs revealed that pHluo_M153R-CD63 is exclusively detected in the exosome-enriched small EV preparation, and not in the larger EVs (Fig. 1d). NTA showed an increased secretion rate of small EVs from pHluo_M153R-CD63-expressing cells compared with parental HT1080s but no change in the number of large EVs (Supplementary Fig. 1a). Live imaging of HT1080 cells stably expressing pHluo_M153R-CD63 as well as the plasma membrane marker mCherry-CaaX revealed numerous pHluo_M153R-CD63-positive puncta left behind migrating HT1080 cells. These puncta were mCherry-CaaX-negative, suggesting that the deposits are likely to be exosomes and not plasma membrane-derived MVs or debris (Fig. 1e and Supplementary Movie 1, upper panel). These findings are similar to the previous green fluorescent slime trails, that we observed left behind cells transiently transfected with pHluorin-CD63[20] (Fig. 1f); however, the deposited trails were much brighter and more easily resolved into puncta using standard epifluorescence imaging (Fig. 1e and Supplementary Movie 1. Note that pHluo-CD63 fluorescence in Fig. 1f and the lower panel of the movie is much dimmer). Also, Western blots of lysates from cells transiently transfected with either pHluorin-CD63[20] or pHluo_M153R-CD63 revealed that our previous construct is present at lower levels than pHluo_M153R. These data suggest that pHluo_M153R-CD63 is indeed more stable (Supplementary Fig. 1b, arrows). Consistent with that idea, we find that the new

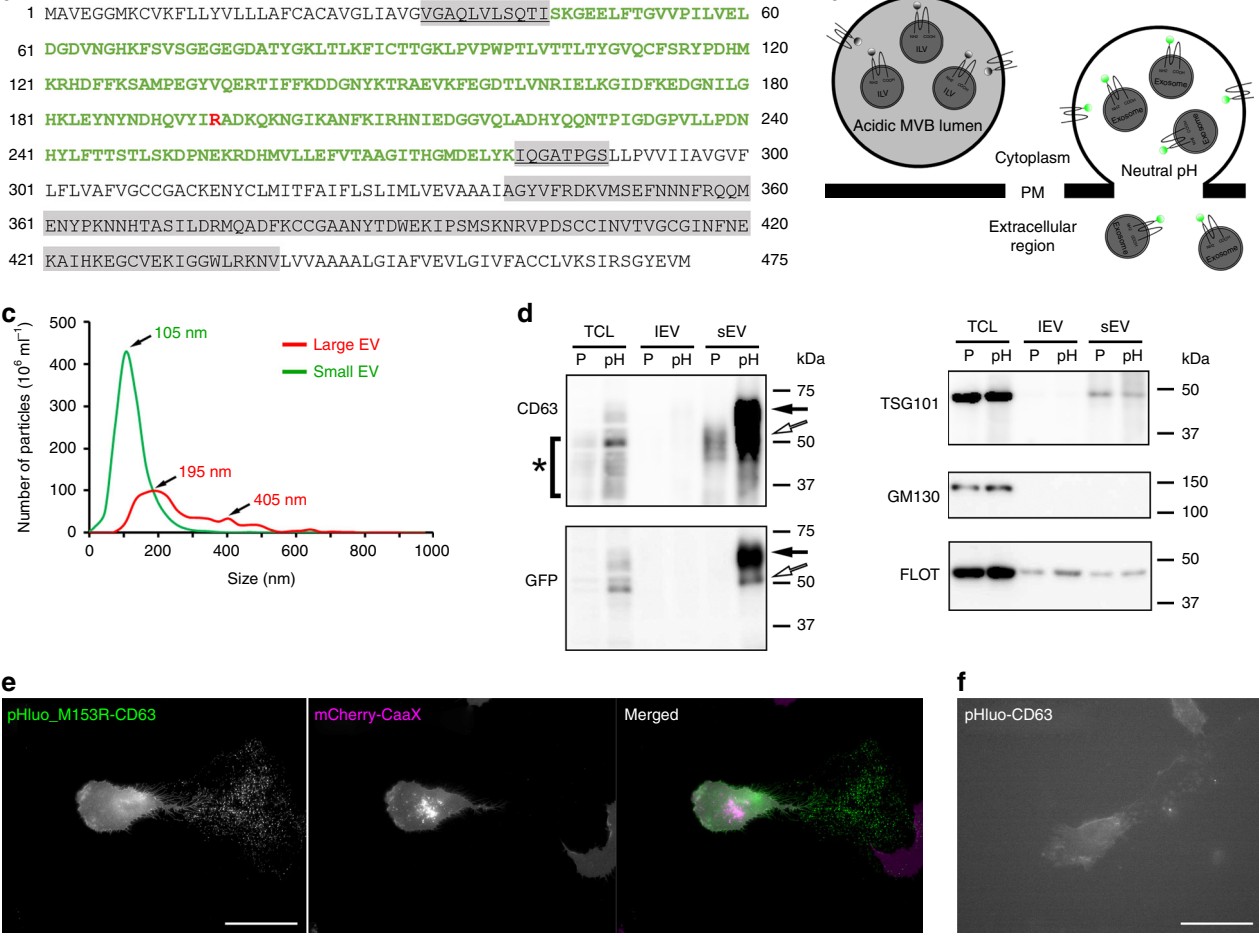

**Fig. 1 pHluo_M153R-CD63 is a bright, stable exosome reporter. a** Sequence of pHluo_M153R-CD63. pHluorin sequence is in green color. Highlighted regions in gray represent small (underlined) and large extracellular loops. M153R mutation is marked in red. **b** Diagram of pHluo_M153R-CD63 construct. Notice pHluo_M153R tag has bright fluorescence upon fusion of the multivesicular body (MVB) with the plasma membrane due to the exposure to neutral pH. Otherwise, it is nonfluorescent in the acidic condition of the MVB lumen. ILV intraluminal vesicle, PM plasma membrane. **c** Representative trace from nanoparticle tracking analysis of large EVs and small EVs. **d** Western blot analysis of cells and EVs with anti-CD63, anti-GFP, EV markers (TSG101, Flotillin) and Golgi marker (GM130). TCL total cell lysate, lEV large EV, sEV small EV, P parental cells, pH pHluo_M153R-CD63-expressing cells. Black arrows indicate full-length pHluo_M153R-tagged CD63, which is shifted due to the GFP moiety of 27 kDa, while white arrows indicate potential cleaved form of CD63 tagged with pHluo_M153R. Asterisk (*) indicates cellular CD63, which has a broad range due to glycosylation. **e** Still images from Supplementary Movie 1 (upper panel) showing a migrating HT1080 cell stably expressing mCherry-CaaX (magenta) and pHluo_M153R-CD63 (green). Colocalization of magenta and green is white. Notice that the deposits left behind the migrating cell are only labeled with CD63 not with CaaX. Scale bar, 50 μm. Movie representative of 38 movies. **f** Still images from Supplementary Movie 1 (lower panel) showing a migrating HT1080 cell transiently expressing pHluo-CD63. Scale bar, 50 μm. Movie representative of 12 movies. Source data are provided as a Source Data file.

reporter can be stably expressed in cells using lentiviral transduction, which has many advantages, including the ability to use flow cytometry to sort cell populations for more uniform fluorescent expression and to image in many more conditions, potentially including low light, lower resolution, 3D and in vivo.

**pHluo_M153R-CD63 is expressed at the surface of small EVs.** Although pellets from $100,000 \times g$ ultracentrifugation contain small EVs, they may be contaminated by nonspecifically bound contaminants, such as protein aggregates. Pellets from $100,000 \times g$ ultracentrifugation were therefore further purified through iodixanol density gradient centrifugation. NTA of the density-gradient-purified small EVs showed the expected size distribution for exosomes with a more narrow distribution and a peak diameter of 135 nm compared to the more broad distribution and peak diameter of 165 nm for large EVs (Fig. 2a). Immunogold transmission electron microscopy (TEM) using an anti-GFP

antibody showed that the pHluorin tag localizes at the surface of density-gradient-purified small EVs (Fig. 2b). Consistent with the well-known heterogeneity of exosomes, not all small EVs purified from HT1080 cells stably expressing pHluo_M153R-CD63 are labeled with the pHluorin tag. In immunoblots, pHluo_M153R-CD63 colocalized with classical EV markers, including CD63, flotillin and HSP70. As expected, the preparations were negative for the Golgi marker GM130 (Fig. 2c).

In a previous study, correlative light-transmission electron microscopy imaging revealed that fluorescent flashes of pHluorin-CD63 at the plasma membrane indeed correspond to MVB fusion events[21]. To determine whether pHluo_M153R-CD63 likewise reports exosome secretion, we knocked down the MVB docking protein Rab27a[23] with shRNA in pHluo_M153R-CD63-expressing HT1080 cells (Fig. 2d). As expected, the number of small EVs released into the media of Rab27a-KD cells was greatly decreased compared to control cells, as assessed by NTA (Fig. 2e). While not all small EVs are expected to be exosomes[24–26], a substantial portion

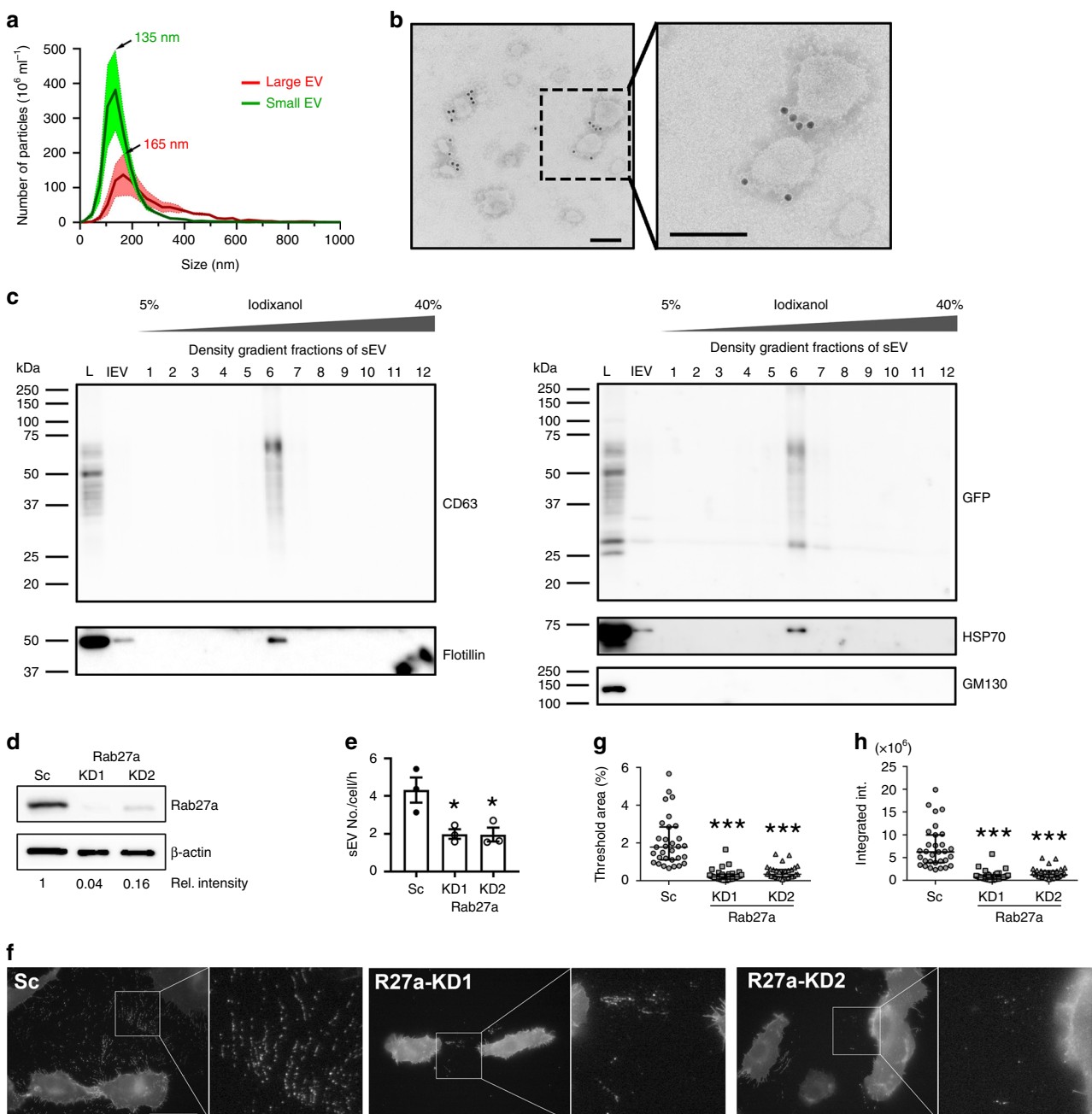

**Fig. 2 pHluo_M153R-CD63 labels exosomes. a** Averaged trace from nanoparticle tracking analyses of large EVs (red) and small EVs (green) secreted from pHluo_M153R-CD63-expressing HT1080. Shaded area represents s.e.m. from three independent experiments. Small EVs were purified through iodixanol density gradient. **b** Representative immunogold negative stain electron micrograph of small EVs iodixanol density-gradient-purified from pHluo_M153R-CD63-expressing HT1080 from two independent experiments. Scale bars, 100 nm. **c** Western blot analysis of cells and EVs with anti-CD63, anti-GFP, EV markers (Flotillin, HSP70) and Golgi marker (GM130). L total cell lysate, lEV large EV, sEV small EV. **d** Immunoblotting of Rab27a in pHluo_M153R-CD63-stably expressing HT1080. Sc scrambled control, KD knockdown. **e** Comparison of small EV (sEV) secretion rate between control and Rab27a-KDs shown as a scatter-plot (mean ± standard error) from three independent experiments. *$P = 0.030793$ and 0.031077 for Rab27a-KD1 and KD2, respectively, by two-sided paired Student's $t$ test. **f–h** Live imaging and analysis of extracellular pHluo_M153R-CD63 for control and Rab27a-KD cells. **f** Representative images, $n = 31$ images from three independent experiments for each cell line. R27a, Rab27a. Scale bar, 30 μm. **g** Threshold area of the extracellular region from (**f**) shown as a scatter-plot (median and quartile range). ***$P < 0.001$ by two-sided unpaired Mann−Whitney $U$ test. **h** Integrated intensity of the extracellular region from (**f**) shown as a scatter-plot (median and quartile range). ***$P < 0.001$ by two-sided unpaired Mann−Whitney $U$ test. Source data are provided as a Source Data file.

of them are expected to derive from MVBs. Imaging of control and Rab27a-KD cells expressing pHluo_M153R-CD63 revealed greatly reduced extracellular deposition of pHluo_M153R-CD63-positive puncta by Rab27a-KD cells (Fig. 2f), as quantitated by measuring the area and integrated intensity of the pHluo_M153R-CD63-positive

deposits surrounding the cells (Fig. 2g, h). pHluo_M153R-CD63-positive deposition also was observed from other cell types (Supplementary Fig. 2). Some of the puncta are brighter than others, suggesting that they may represent groups of exosomes. Furthermore, some puncta are arranged in linear trails, which may

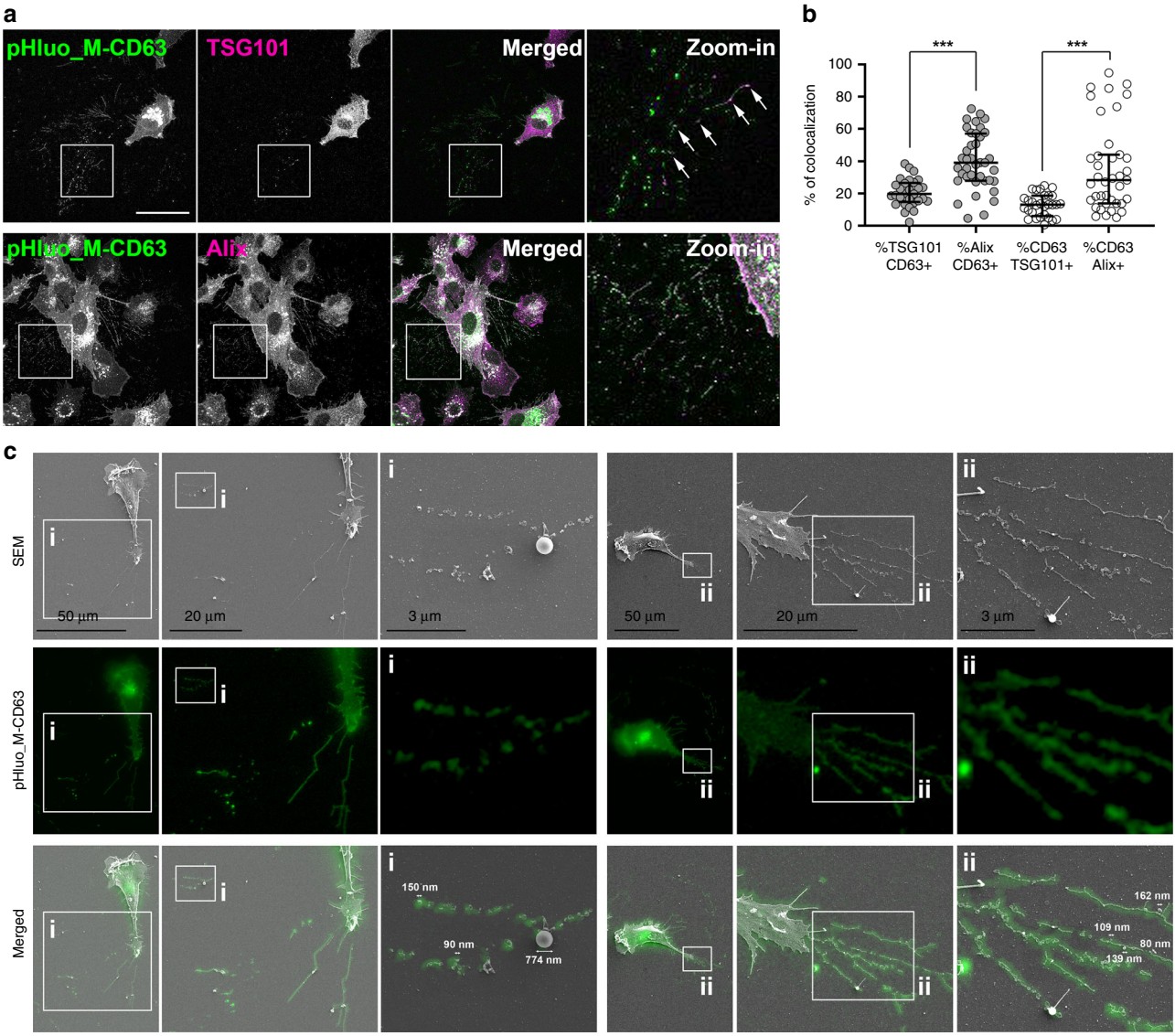

**Fig. 3 pHluo_M153R-CD63-positive trails mark secreted exosomes. a** Representative images of immunofluorescent staining of fixed cells for the exosomal markers TSG101 or Alix (magenta) and pHluorin (green). Arrows indicate colocalization of TSG101 (magenta) with pHluo_M153R-CD63-positive puncta (green) to form white. Note many white puncta in the CD63 + Alix merged images. pHluo_M-CD63, pHluorin_M153R-CD63. $n = 29$ and 39 images for TSG101 and Alix, respectively, from four independent experiments. Scale bar, 30 μm. **b** Colocalization analysis from (**a**) shown as a scatter-plot (median and quartile range). ***$P < 0.001$ by two-sided unpaired Mann−Whitney $U$ test. **c** Correlative light-electron microscopy of pHluo_M153R-CD63-stably expressing HT1080 (green). Increasing magnification of images is shown to the right and indicated with i or ii. Representative of 22 cells from four experiments. SEM scanning electron microscopy. Source data are provided as a Source Data file.

represent organization by cells as they migrate over them (e.g. as in Supplementary Movie 1, trailing edge of migrating cell).

**Extracellular pHluo_M153R-CD63 puncta are exosome deposits.** As a further confirmation, we colocalized the pHluo_M153R-CD63-positive extracellular puncta with additional exosome markers, specifically those associated with the ESCRT machinery that causes intraluminal vesiculation within MVB (Fig. 3a). Immunostaining of fixed cells revealed that ~20% of extracellular puncta that are positive for the ESCRT-I protein TSG101 are also positive for pHluo_M153R-CD63. Immunostaining with the ESCRT accessory protein, Alix, revealed even more colocalization with pHluo_M153R-CD63 in extracellular puncta (median ~40%, Fig. 3b). Analysis of the percent pHluo_M153R-CD63-positive area that is positive for TSG101 or Alix likewise revealed

more colocalization of pHluorin with Alix than TSG101, in some images almost 100% colocalization. The difference in colocalization with TSG101 versus Alix is consistent with the well-known heterogeneity of exosomes as well as with previous immunofractionation experiments in which an exosome biogenesis syndecan-syntenin-Alix complex was present in EVs immunoprecipitated with anti-CD63 antibody[10]. Of note, due to neutralization of intracellular pH by the paraformaldehyde fixation procedure, pHluo_M153R-CD63 shows bright fluorescence in internal endosomal structures in these images (Fig. 3a).

To further determine whether pHluo_M153R-CD63 extracellular puncta and trails represent exosome deposition, we performed correlative light-electron microscopy in which HT1080 cells expressing pHluo_M153R-CD63 were first observed by epifluorescence microscopy followed by scanning electron microscopy (Fig. 3c). pHluo_M153R-CD63 fluorescence corresponded to small

EVs with diameters ranging from 80 to 160 nm, which were frequently organized into linear trails (Fig. 3c-i) and/or associated with retraction fibers (Fig. 3c-ii). By contrast, larger vesicles (e.g. the 774 nm EV in Fig. 3c-i) were occasionally observed and were not fluorescent. These data are consistent with CD63 labeling primarily exosomes and not larger MVs[20,27] (Fig. 2c). We also observed rare pouches of small vesicles left behind cells (Supplementary Fig. 3). These pouches resemble previously described migrasomes[28], which are reportedly driven by different tetraspanins[29]. Indeed, the vesicles inside the pouches do not appear to be fluorescent, suggesting that the vesicles may be of a different origin.

**Cells migrate toward extracellular pHluo_M153R-CD63 deposits.** Previous reports indicate that exosome secretion promotes directional migration of several cell types including cancer cells[20,30], neutrophils[31] and Dictyostelia[32]. To visualize dynamically how secreted exosomes influence cell migration, we performed live imaging under diverse conditions. In two-dimensions, mCherry-CaaX/pHluo_M153R-CD63-double-labeled HT1080 cells exhibited pathfinding behavior along exosomes. Thus, cells extended leading edge protrusions toward and over exosome deposits labeled with pHluo_M153R-CD63 (Fig. 4a, arrows, Supplementary Movie 2) and then migrated along the deposits (Fig. 4a, arrowheads). We observed the same phenotype in 3D collagen gels (Fig. 4c, Supplementary Movie 3). To provide a quantitative assessment of this behavior, we analyzed the relationship of cell migration paths relative to exosomes by measuring the angle of each cell trajectory to the nearest exosome trail. We then took the cosine of the angle to obtain the pathfinding index. Thus, zero deviation of the path from an exosome trail is 1, i.e. if cells migrated directly toward the nearest exosome deposits or if cells migrated directly over the deposits. If cells migrated away from the nearest deposits in the opposite direction with a 180° angle, then the pathfinding index would be −1, as shown in the cartoons in Fig. 4b, d. For both 2D and 3D migration, HT1080 cells primarily migrated toward or over exosome deposits with a median pathfinding index of 1 and 0.0607 and 0.0254 quartile ranges for 2D and 3D migration, respectively (Fig. 4b, d).

To further examine whether pHluo_M153R-CD63 extracellular puncta are visible in living systems, we performed intravital imaging in mice as well as in chick embryos. For the mouse system, MDA-MB-231 breast cancer cells stably expressing pHluo_M153R-CD63 were injected into the fourth mammary fat pad and observed 7−10 days later by multiphoton imaging through a mammary imaging window (Fig. 4e). Simultaneous with the GFP imaging, collagen fibers were imaged using second harmonic generation[33]. For the chick embryo, HT1080 fibrosarcoma cells expressing pHluo_M153R-CD63/mCherry-CaaX were injected into the chorioallantoic vein and imaged after 24 h using an upright epifluorescence microscope (Fig. 4f). In both systems, bright fluorescent pHluo_M153R-CD63-positive puncta were observed surrounding the pHluo_M153R-CD63-expressing cancer cells (Fig. 4e, f, arrowheads). Similar to our in vitro observations, these puncta were frequently associated with long cellular extensions that could be either retraction fibers or filopodia (Fig. 4e, f, Supplementary Movie 4, arrows). In the mouse xenograft system, some exosome deposits appeared to adhere to collagen fibers (Fig. 4e, dotted circles). These extracellular exosome deposits could theoretically provide a substrate for cell migration as we previously proposed[20], and now observe in 2D and 3D (Fig. 4a–d).

Finally, to characterize the in vivo stability of pHluo_M153R-CD63-positive EVs, we measured their half-life in circulation by injecting small EVs purified from HT1080-pHluo_M153R-CD63 cells into the chorioallantoic vein of chick embryos. Plasma was collected both before and immediately after injection, as well as at 30-min intervals thereafter. The presence of pHluorin-positive EVs at various time points was assessed by EV flow cytometry, using the CellStream flow cytometer (see Methods and Supplementary Fig. 4). The half-life of the pHluorin-positive EVs in plasma was calculated from the decay curve and found to be ~15 min (Fig. 4g). These data are consistent with rapid removal of EVs from the circulation, as previously identified in refs. [34,35].

**Exosome deposits occur at the front of migrating cells.** The previously identified role of exosomes in directional migration[20,30–32] suggests that exosomes should be secreted from the front of migrating cells. However, it has been difficult under standard conditions to definitively determine whether exosomes are secreted at the front or back (or both) of migrating cells, especially since CD63 localizes to both the plasma membrane and exosomes. To more definitively answer this question, we seeded pHluo_M153R-CD63-expressing HT1080 cells on dishes with a polymeric nanopatterned surface topography and performed confocal microscopy of migrating cells (Fig. 5a, Supplementary Movie 5). Due to the patterning of the substrate, new pHluo_M153R-CD63 deposits were easy to observe and track. From these movies, we observed that new exosome deposits (depicted with cyan color in the heat map) clearly occurred at the front of the cell and were largely stationary as the cell moved over them, with reorganization of the deposits at the rear of the cell by retraction fibers. Using a criterion of <2 μm of displacement length during the movie as immotile, we find that 85% of deposits are immotile (Fig. 5b, c). While we cannot rule out the possibility that secretion also occurs from the rear of the cell (this is difficult to assess due the accumulation of fluorescence at the rear of the moving cell, presumably from ongoing secretion), some secretion clearly occurs at the front of the cell and thus could mediate the cellular directional sensing that has been shown to be a critical function of exosome secretion[20,30,31]. We should also note that we do not observe any apically secreted exosomes in this assay, likely due to both the plane of focus of the microscope and the fact that any apically secreted exosomes would be likely to rapidly float away.

**Creation of a dual-color reporter to track MVBs.** We previously observed by TIRF microscopy that putative MVB fusion events closely precede adhesion formation[20] in spreading cells. However, our ability to track and observe fusion events in other contexts (e.g. by widefield or confocal microscopy) was limited due to the dim fluorescence of the original pHluorin-CD63 construct, and lack of a pH-insensitive tag to track MVB trafficking events before fusion with the plasma membrane. To solve the latter problem, mScarlet, a bright monomeric red fluorescent protein[36], was cloned at the C-terminus of pHluo_M153R-CD63 to make pHluo_M153R-CD63-mScarlet (Supplementary Fig. 5a). This construct exhibits red fluorescence under acidic conditions (e.g. when contained in the MVB lumen) and dual fluorescence under neutral conditions (e.g. after secretion) (Supplementary Fig. 5b). Density-gradient-purified small EVs secreted from HT1080 stably expressing pHluo_M153R-CD63-mScarlet had the expected size distribution with a peak of 135 nm while large EVs have peaks of 195 and 345 nm (Supplementary Fig. 5c). pHluorin tag was detected at the surface of exosomes by immunogold TEM using anti-GFP antibody (Supplementary Fig. 5d). As there are no commercially available antibodies to mScarlet, we were not able to check its localization. In addition, pHluo_M153R-CD63-mScarlet cofractionated with EV markers in density gradients (Supplementary Fig. 5e) and was detected at a higher molecular weight than pHluo_M153R-CD63 (compare to Fig. 2c).

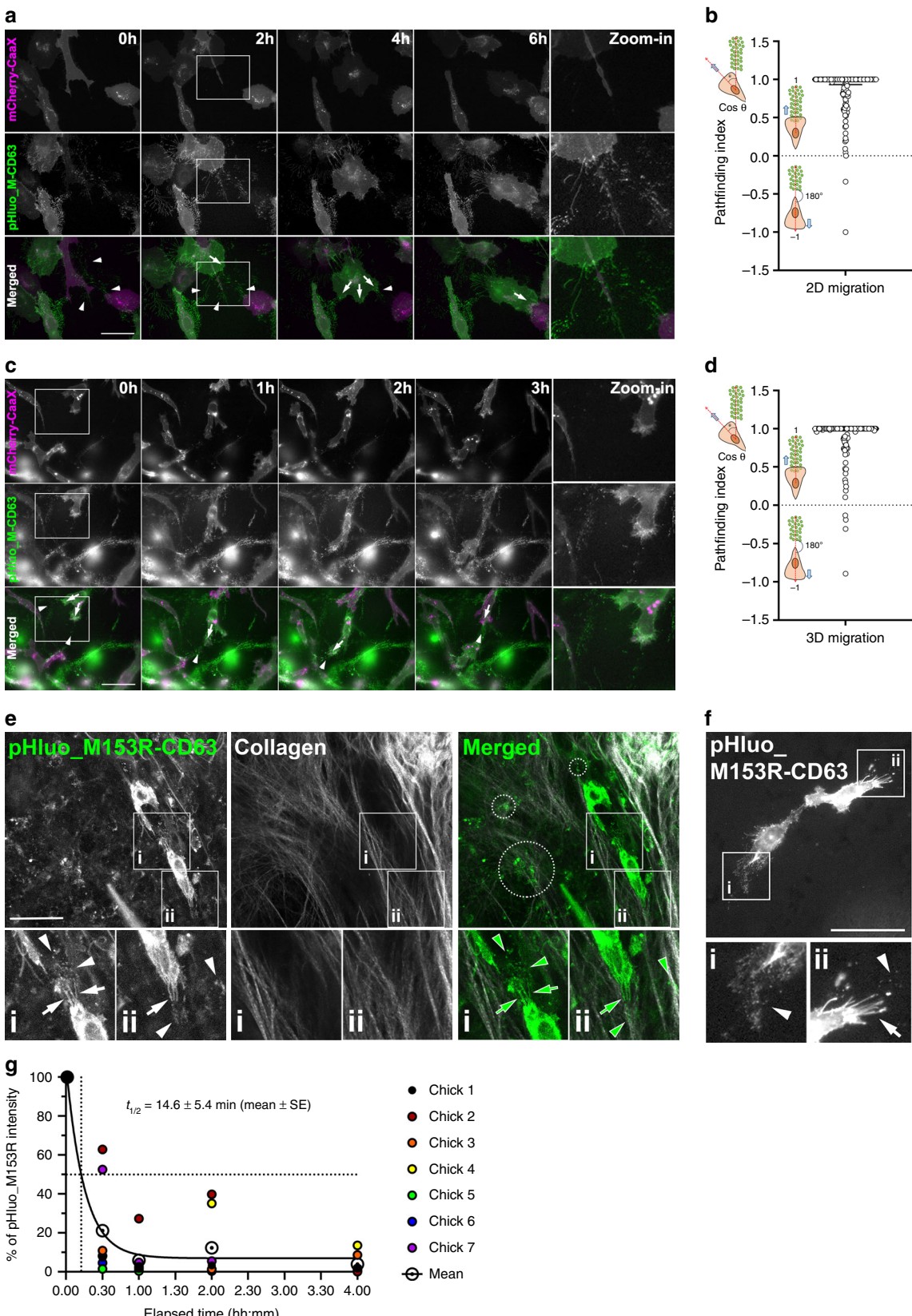

The C-terminal cytosolic tail sequence GYEVM is the lysosomal targeting motif of CD63[37]. To avoid interference of the mScarlet tag with endolysosomal targeting, we included two extra amino acids (glutamic acid and phenylalanine, Supplementary Fig. 5a, bolded and underlined) between the C-terminus of CD63 and the N-terminus of mScarlet. To determine whether the C-terminal mScarlet tag interferes with CD63 trafficking, we performed immunofluorescence for several late endosomal/lysosomal markers and analyzed the effect of the tag on colocalization (Fig. 6). Comparing the localization of

**Fig. 4 Cells exhibit pathfinding behavior on exosome trails. a** Time-series images from live imaging of pHluo_M153R-CD63 (green)/mCherry-CaaX (magenta) dual-expressing HT1080 cells in 2D culture conditions (see Supplementary Movie 2). mCherry-CaaX is shown in magenta and pHluo_M153R-CD63 is shown in green in the merged image. *n* = 186 cells from 31 movies from four independent experiments. Arrowheads indicate exosome trails. Arrows indicate protrusions. pHluo_M-CD63, pHluo_M153R-CD63. Scale bar, 50 μm. **b** Pathfinding index from (**a**) shown as a scatter-plot with median and quartile range. **c** Time-series images from live imaging of pHluo_M153R-CD63 (green)/mCherry-CaaX (magenta) dual-expressing HT1080 cells in 3D collagen gels (see Supplementary Movie 3). *n* = 151 cells from 27 movies from four independent experiments. Arrowheads indicate exosome trails. Arrows indicate protrusions. Scale bar, 50 μm. **d** Pathfinding index from (**c**) shown as a scatter-plot with median and quartile range. **e** Representative intravital images of pHluo_M153R-CD63 (green in merge) and collagen fibers (white in merge) in mice from eight independent experiments. Multiphoton *z*-stacks shown as maximum intensity-projections. Arrows indicate adhesive protrusions. Arrowheads indicate exosome deposits. Examples of exosome deposits associated with collagen fibers are shown in dotted circles. Scale bar, 50 μm. **f** Representative intravital images of pHluo_M153R-CD63 in chicks from six independent experiments. Arrows indicate adhesive protrusions. Arrowheads indicate exosome deposits. Scale bar, 50 μm. **g** Half-life determination of pHluo_M153R-CD63 small EVs in vivo using the avian embryo model (*n* = 7 chick embryos from three independent experiments). Flow cytometry was performed on chick plasma collected at each time point upon intravenous injection of pHluo_M153R-CD63 small EVs. Gated pHluo_M153-positive events (pHluo_M153R+) at each time were used for half-life calculations in GraphPad Prism. The gating strategy is shown in Supplementary Fig. 4. Source data are provided as a Source Data file.

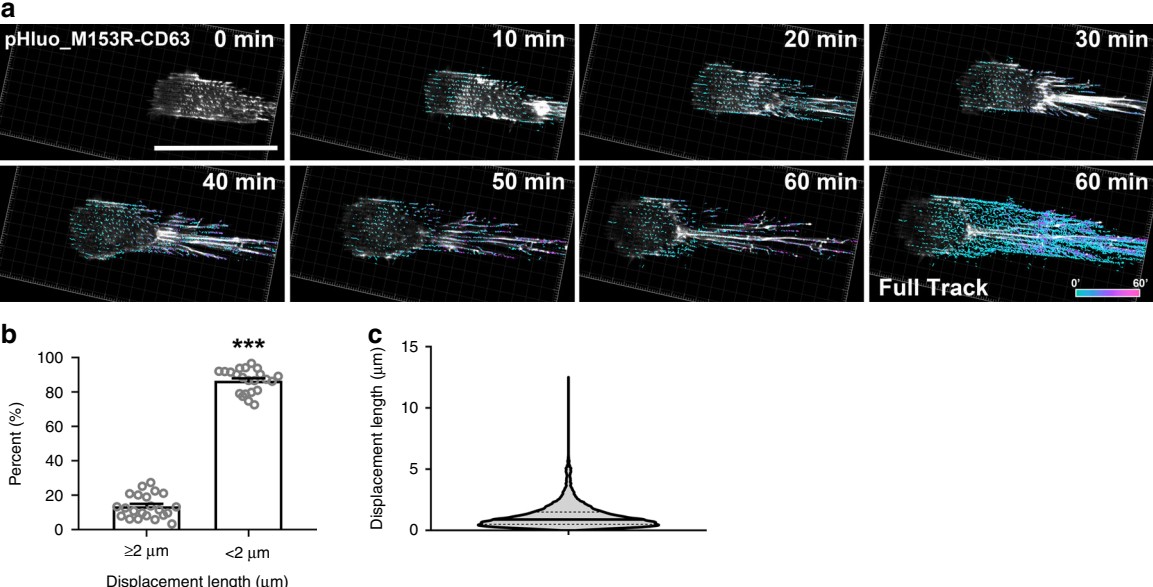

**Fig. 5 Exosomes are deposited at the front of migrating cells on a nanopatterned dish. a** Time series from Supplementary Movie 5, showing trajectories of tracked exosome deposits under the cell on a nanopatterned dish. Colors on heat map represent the time elapsed since deposition occurred. Note that cyan color shows newly deposited exosomes. Note the image at the right bottom corner (Full Track) shows the overlap of all of the tracks from the movie. *n* = 23 cells from four independent experiments. Scale bar, 50 μm. **b, c** Analysis of mobility of exosome deposits from movies as in (**a**) with scatter dot plots in (**b**) (mean ± standard error) showing the percent of exosome deposits/cell with displacement length > or <2 μm, where each dot represents the median displacement length of exosome deposits in that category from each single cell. ***P < 0.001 by two-sided paired Student's *t* test. **c** Total events (*n* = 34,052 from 23 **c**ells) of displacement length are shown in a violin plot (median with interquartile range) from (**a**). Source data are provided as a Source Data file.

pHluo_M153R-CD63 and pHluo_M153R-CD63-mScarlet in stably expressing HT1080 cells revealed that ∼ 55% of the lysosomal marker LAMP1 is colocalized with pHluorin from HT1080-pHluo_M153R-CD63 cells (Fig. 6a, b). Similar colocalization was observed with either dual fluorescence or pHluorin from pHluo_M153R-CD63-mScarlet cells (55−60% colocalization with LAMP1). A small but significant increase in colocalization of mScarlet-only fluorescence with LAMP1 (∼66.9%) was observed in HT1080-pHluo_M153R-CD63-mScarlet cells (Fig. 6b). However, LAMP1 colocalization with pHluorin from either the single or dual tagged cells is not significantly different, suggesting that mScarlet tag at the C-terminus of CD63 does not interfere with CD63 targeting to lysosomes. The higher rate of colocalization of LAMP1 with the red signal may be due to higher susceptibility of GFP than mScarlet to lysosomal acidic and degradative conditions, such as that occurs with tandem fluorescent-tagged LC3 (mRFP-EGFP-

LC3)[38]. We also find no significant difference between colocalization of endogenous CD63 or pHluo_M153R-CD63-mScarlet (dual or single fluorescence) with the late endosomal marker Rab7 (Fig. 6c, d).

We also analyzed whether the GFP and mScarlet signals from pHluo_M153R-CD63-mScarlet colocalize with CD63 immunostaining in extracellular EV deposits. Indeed, we find that ∼80% of extracellular mScarlet and pHluorin puncta from HT1080-pHluo_M153R-CD63-mScarlet cells colocalize with each other and with total extracellular CD63 puncta, which includes both tagged and endogenously produced CD63 (Fig. 6e–g). While the colocalization of tag fluorescence with CD63 was similar (Dual; 86.5 ± 6.4%, mScarlet; 83.2 ± 5.3%, pHluorin; 79.4 ± 7.0%, all mean ± SD), there was a small but significant increase in the colocalization of the dual tag with CD63 compared to the single tags. This difference might reflect a small amount of degradation that is more apparent with the single fluorescent channels.

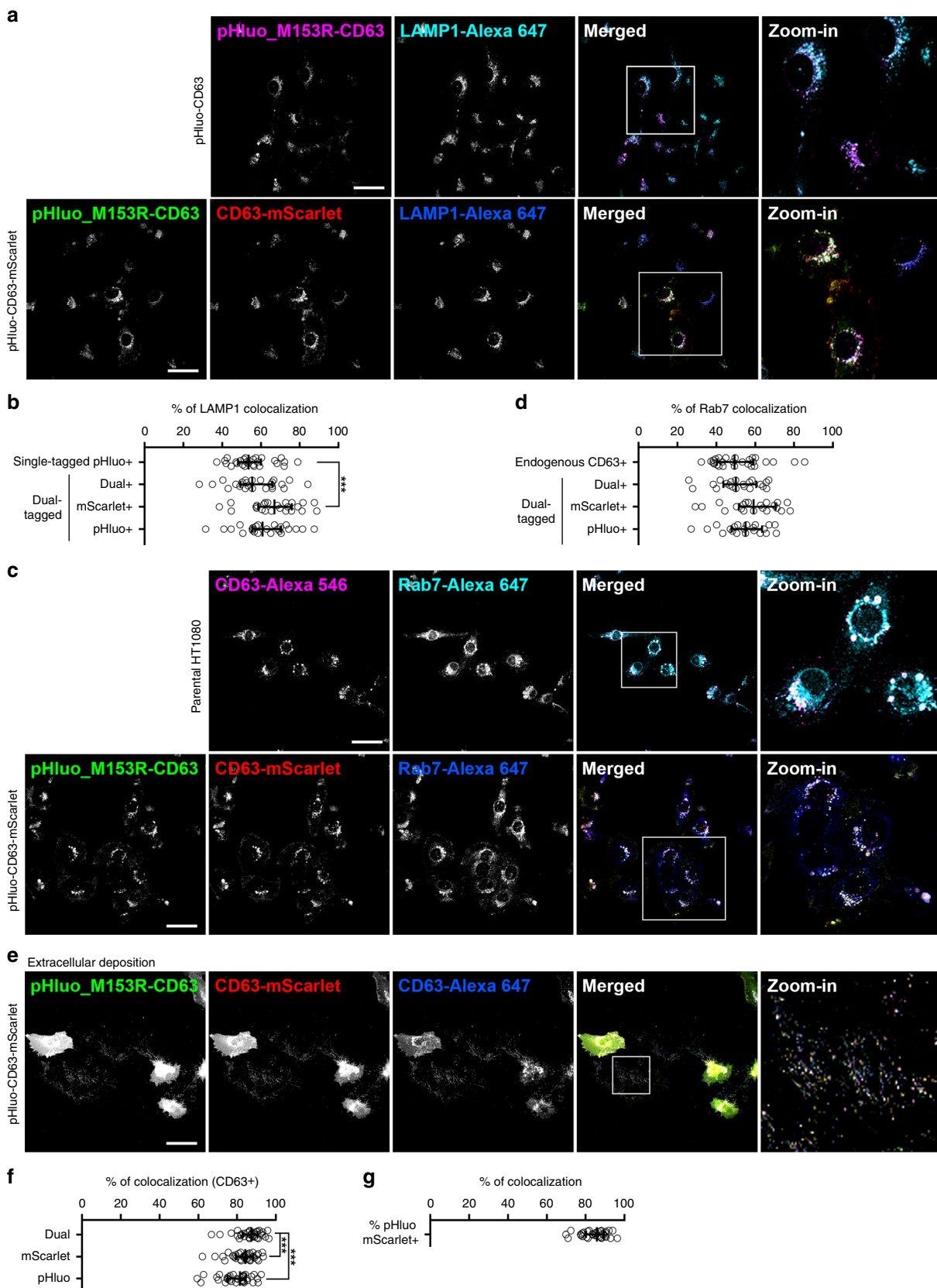

To test the functionality of pHluo_M153R-CD63-mScarlet for observing MVB trafficking and fusion, live confocal microscopy of HT1080 cells stably expressing pHluo_M153R-CD63-mScarlet was performed. The movies revealed that MVBs are frequently trafficked to protruding cell edges and fuse there (Fig. 7a and Supplementary Movie 6). Note the frequent appearance of pHluo_M153R-CD63-mScarlet, often before membrane protrusion.

**Fig. 6 mScarlet tag does not interfere with localization or function of CD63. a** Representative images of immunofluorescent staining of LAMP1 (cyan or blue) in HT1080-pHluo_M153R-CD63 cells (n = 210 cells from 28 images from three independent experiments) and HT1080-pHluo-M153R-CD63-mScarlet cells (n = 313 cells from 24 images from three independent experiments). pHluo_M153R-CD63 from a single tag and a dual tag is shown in magenta (the upper raw) and green (the lower raw), respectively. CD63-mScarlet is shown in red. Scale bars, 50 μm. **b** Percentage of LAMP1 colocalized with tagged fluorescent proteins from (**a**). Data are presented as median with interquartile range. ***P = 0.0009 by two-sided unpaired Student's t test. **c** Representative images of immunofluorescent staining of Rab7 in parental HT1080 cells (n = 343 cells from 27 images from four independent experiments) and HT1080-pHluo_M153R-CD63-mScarlet cells (n = 255 cells from 21 images from three independent experiments). Rab7 is shown in cyan (the upper raw) or in blue (the lower raw). Endogenous CD63 from parental HT1080 is shown in magenta. pHluo_M153R-CD63 and CD63-mScarlet are shown in green and red, respectively. Scale bars, 50 μm. **d** Percentage of Rab7 colocalized with endogenous CD63 and tagged fluorescent proteins from (**c**). Data are presented with median with interquartile range. Statistics was tested by two-sided unpaired Student's t test. **e** Representative images of immunofluorescent staining of CD63 (blue) for extracellular deposits from HT1080-pHluo_M153R-CD63-mScarlet cells (n = 34 images from three independent experiments). pHluo_M153R-CD63 and CD63-mScarlet are shown in green and red, respectively. Scale bar, 50 μm. **f** Percentage of tagged fluorescent proteins that colocalized with CD63 from (**e**). Data are presented with median with interquartile range. ***P < 0.001 by two-sided paired Wilcoxon test. **g** Percentage of pHluo_M153R colocalized with mScarlet from (**e**). Data are presented with median with interquartile range. Source data are provided as a Source Data file.

It was previously shown by DIC and widefield epifluorescence microscopy that exosomes can be captured by filopodia and endocytosed into cells[39]. Using the dual-color reporter, we observed a similar event in which migrating cells detect exosome deposits by touching them with filopodia, then migrate toward the exosome deposits and engulf them (Fig. 7b, white arrows, Supplementary Movie 7). Although these experiments were not designed to identify mechanisms of endocytosis[40], in some cases we observed micropinocytosis-like uptake by engulfment (25 min frame, top arrow). Of note, endocytosed exosomes turned from yellow (pseudocolored white) to red (pseudocolored magenta) over time, consistent with maturation and acidification of endosomes (Fig. 7b, magenta arrows). From the time-lapse images, we quantified the kinetics of internalization and acidification of endocytosed exosome deposits. The internalization time of exosome deposits from the initial filopodial contact was highly variable and ranged from 1 to 20 min with a median internalization time of 6 min (Fig. 7c). The distance between the exosome deposits and the plasma membrane likely affects the length of time taken for exosome uptake after initial contact. However, acidification of endocytosed exosome deposits, as visualized by loss of the pHluorin signal, occurred in a narrower time range, from 2 to 11 min with a median of 7 min (Fig. 7d).

## Discussion

Dynamic monitoring of EVs is critical to investigate the roles of EV secretion in cell behaviors, including cell–cell communication and cell migration. Although we previously developed a pH-sensitive exosome secretion reporter, pHluorin-CD63[20], the reporter was dim and could not be stably expressed in cells. By making a single amino acid mutation, we developed a stable and bright pH-sensitive reporter, pHluo_M153R-CD63, for live imaging of MVB fusion and cellular interactions with extracellular exosomes. Using this reporter, we observed that exosomes are secreted at the front of migrating cells and left behind in exosome trails. We also observed strong pathfinding behavior of migrating cells along trails of extracellular exosomes. Finally, we made a dual-color reporter that allows observation of not only exosome secretion but also internal trafficking events in both donor and recipient cells.

Recently, there has been much interest in observing EV release and interchange between cells. GFP-CD63 and fluorophores tagged to palmitoylation motifs have been used to observe uptake by recipient cells within tumors[14,41]. In zebrafish, recent papers used both pHluorin-CD63 (injected as plasmid DNA for transient expression) and a cyanine-based membrane probe to observe EV dynamics in the blood circulation[35,42]. Our new reporters provide important additions to these tools by allowing stable expression in diverse cell types and potentially in transgenic animals while allowing high-resolution imaging of subcellular events. As with all

reporters, overexpression of a protein may lead to boosting of that protein's functions in cells. We did find that the pHluo_M153R-CD63-overexpressing cells secreted more small EVs into conditioned medium, consistent with the reported role for CD63 in exosome biogenesis[13]. Possible ways to avoid any perturbations include expressing at lower levels or tagging the endogenous gene with pHluo_M153R using gene editing.

A key question addressed in this study is whether pHluo_M153R-CD63-positive extracellular puncta correspond to exosomes. As shown by Western blot analysis and CLEM, pHluo_M153R-CD63 labels small but not large EVs. The extracellular pHluorin-positive puncta were not labeled with the plasma membrane marker, mCherry-CaaX, suggesting that they did not derive from the plasma membrane but rather from internal MVBs. Consistent with that hypothesis, KD of the MVB docking factor Rab27a[23] greatly reduced both the secretion of small EVs and the deposition of pHluo_M153R-CD63-positive extracellular puncta compared to control cells. Of note, there was a larger reduction in the area and intensity of extracellular CD63-positive deposits (Fig. 2f–h) than in the number of secreted small EVs measured by NTA (Fig. 2e). This discrepancy indicates the heterogeneity of small EVs and suggests the possibility of the release of small CD63-negative ectosomes from the plasma membrane[26], as these would not be subject to regulation by Rab27a-mediated MVB docking. We also observed through immunofluorescent staining that pHluo_M153R-CD63-positive puncta are more frequently positive for Alix, an ESCRT accessory protein, than for TSG101, an ESCRT-I protein. These data are consistent with a previous finding that syntenin regulates the budding of CD63-positive ILVs into MVBs by interaction with Alix[10], although TSG101 was also implicated in that process.

Exosome secretion promotes chemotaxis of several cell types[30–32]. Previously, we proposed a model in which exosomes secreted from cancer cells promote cancer cell chemotaxis in both an autocrine manner by secretion at the leading edge as well as in a paracrine manner by leaving behind exosome trails[30]. More recently, the Parent group reported that exosome trails released from migrating Dictyostelia induce streaming behavior[32]. In this study, we observed that cancer cells migrate toward and over exosome deposits to use them as migration tracks in 2D and 3D tissue culture environments. Although it is not clear how much of this behavior is due to chemicals released from the exosomes to induce chemotaxis and how much is providing an adhesive[20] and signaling migration track, through direct interaction, the overall behavior is clear. It is also possible that proteinases associated with exosomes[43], other EVs[44] and/or secreted in a soluble form may create tunnels in the 3D matrix that contribute to the leader–follower behavior; however, this could not account for the highly similar pathfinding behavior along exosome trails that we observe during 2D migration (Fig. 4).

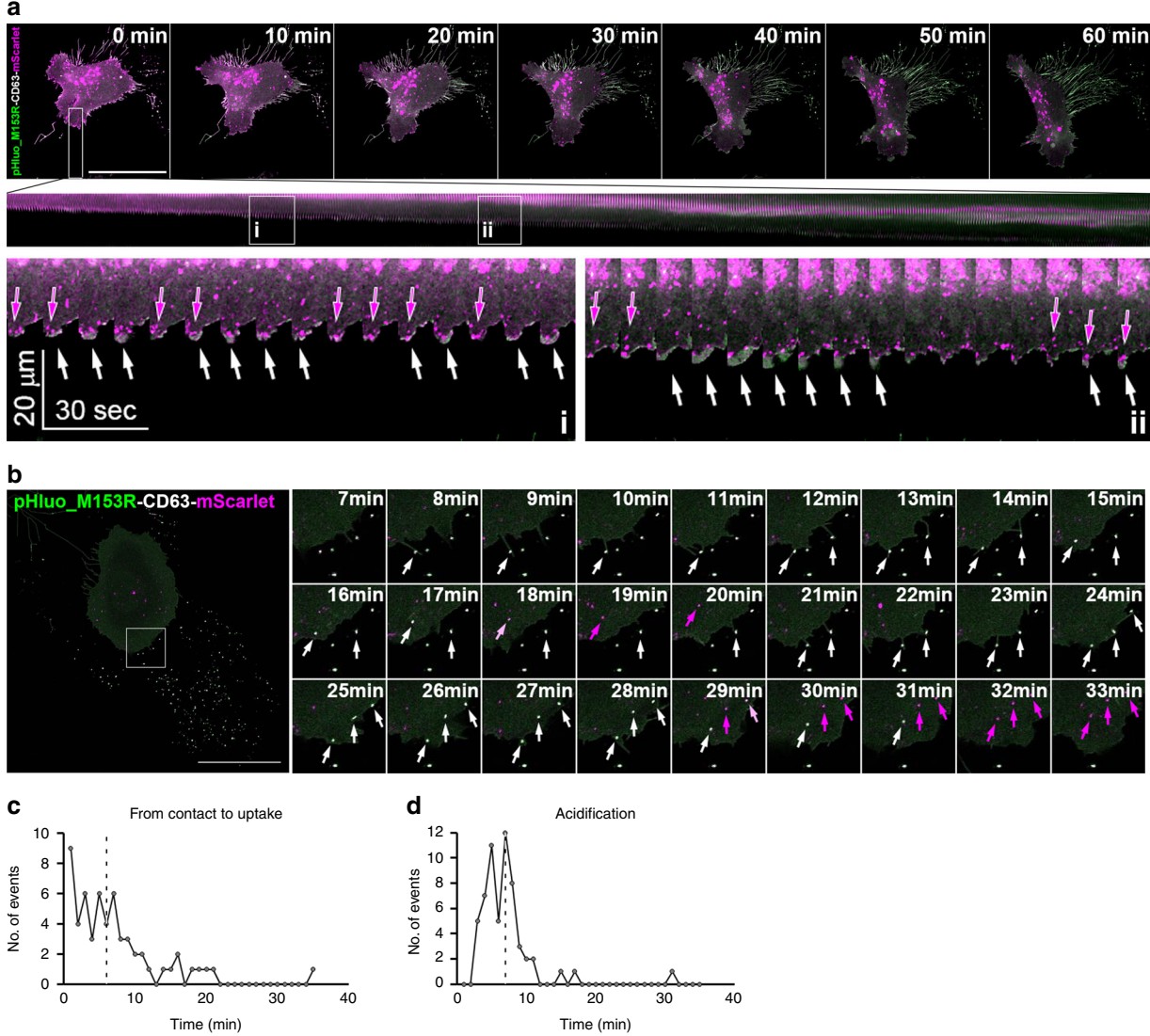

**Fig. 7 Dual reporter reveals MVB transport before fusion and endosome acidification after uptake.** Live confocal microscopy was used to image pHluo_M153R-CD63-mScarlet-expressing HT1080 cells. **a** Representative of 35 cells from four independent experiments. Top: Images were captured at 10 s intervals for 60 min (see Supplementary Movie 6). Time-series images with 10 min intervals show frames at the beginning and end of the movie, with the rectangle across the leading edge in the 0 min frame indicating location from which the kymograph (middle) was derived. pHluo_M153R-CD63 and CD63-mScarlet are shown in green and magenta, respectively. Scale bar, 50 μm. Middle: Kymograph shows events over time across the indicated rectangle. Two regions (i and ii) selected from kymograph (middle) are enlarged at the bottom of the panel. Magenta arrows in kymograph zooms indicate CD63-mScarlet-positive MVB trafficking and docking to protrusions while white arrows indicate MVB fusion and exosome secretion. **b** Time-series images from live confocal microscopy of pHluo_M153R-CD63-mScarlet-expressing HT1080 cells with 1 min intervals from Supplementary Movie 7. White arrows indicate examples where filopodia contact exosome deposits. Magenta arrows indicate acidified exosome-containing endosomal compartments after exosome uptake. pHluo_M153R-CD63 and CD63-mScarlet are shown in green and magenta, respectively. n = 58 exosome uptake events from ten cells from five independent experiments. Scale bar, 25 μm. **c** Quantitation of internalization time of exosome deposits after initial filopodial contact from (**b**). Median time from contact to uptake is marked by a dotted line. **d** Quantitation of time from exosome endocytosis to acidification as shown in (**b**). Median time of acidification is marked by a dotted line. Source data are provided as a Source Data file.

Future studies using our reporter, both in vivo and in vitro, in conjunction with molecular manipulations should be able to dissect further these behaviors.

Another striking finding, observed with our dual-color reporter, is that cancer cells migrate toward exosome deposits and actively endocytose them. Heusermann et al.[39] previously reported that exosomes enter cells through filopodia, a similar behavior to that of viruses[45]. However, for the first time, we visualized and quantitated both exosome uptake and the subsequent acidification of the exosome-containing endosomal compartments. We anticipate that future studies may be able to

use this dual-color reporter to monitor exosome interchange between cells and whether endocytosed exosomes are recycled and re-secreted. Furthermore, by combining with inhibitors, molecular interventions, and/or other reporter systems, the dual reporter may be useful for studying the endocytic fates of exosomes in diverse cell types.

Previous studies have reported that exosomes are likely secreted from the uropod or tail of migrating cells[46,47]; however, those studies were performed with Dictyostelia and leukocytes that primarily use amoeboid locomotion for movement, involving the formation of posterior uropods. By contrast, mesenchymal

migration and directional sensing of cells undergoing chemotaxis would seem to require exosome secretion at the leading edge of cells, at least for single-cell autocrine migration events. Due to the fact that cells migrate over exosome paths, it has been difficult to discern where exosomes are secreted at a subcellular level. Using polymeric nanopatterned dishes, we observed clear deposition of pHluo_M153R-CD63-positive puncta at the front of migrating HT1080 cells. The deposited puncta mostly stayed stationary as the cells moved over them and were left behind the migrating cells, with a small amount of reorganization along the way. We also observed that migrating cells left bright retraction fibers at the trailing edge attached to the exosome deposits. The strong adherence of the cells to the secreted exosomes in both the nanopatterned substrate experiments and in our other movies is consistent with the adhesive function of exosomes that we previously reported[20,30]. Due to the bright accumulation of previously deposited exosomes, we could not observe or rule out additional exosome deposition at the rear of cells migrating on the nanopatterned plates; however, it is clear that exosomes are secreted at the cell front during migration.

To conclude, pHluo_M153R-CD63 is a stable and bright live-cell reporter of exosome secretion. Migrating cells leave pHluo_M153R-CD63-positive exosome trails behind them and the exosome trails attract and promote migration of follower cells. With our dual-color reporter pHluo_M153R-CD63-mScarlet, it is also possible to monitor MVB trafficking before fusion as well as exosome endocytosis. We anticipate that these reporters will be broadly useful to investigate regulation and functions of exosome secretion and uptake in diverse physiological conditions.

## Methods

**Cell culture and reagents**. HT1080 fibrosarcoma cells were maintained in Dulbecco's Modified Eagle Medium (DMEM) supplemented with 10% bovine growth serum. MDA-MB-231 breast cancer cells were maintained in DMEM supplemented with 10% fetal bovine serum (FBS). HNSCC61 head and neck squamous cell carcinoma cells were a gift from Wendell Yarbrough (University of North Carolina at Chapel Hill School of Medicine, Chapel Hill, North Carolina, USA) and maintained in DMEM supplemented with 20% FBS and 0.4 µg ml$^{-1}$ hydrocortisone. A lentiviral shRNA expression system, pLKO.1, was used to knockdown Rab27a (TRCN0000005296 (5′-CCGG-CGGATCAGTTAAGTGAAGAAA-CTC GAG-TTTCTTCACTTAACTGATCCG-TTTTT-3′) and TRCN0000005297 (5′-CCGG-GCTGCCAATGGGACAAACATA-CTCGAG-TATGTTTGTCCCATT GGCAGC-TTTTT-3′, Thermo Fisher Scientific) or scrambled control (5′-CCT AAGGTTAAGTCGCCCTCG-3′, Addgene Plasmid #26701). Viral particles were produced from 293FT cells by cotransfection with viral vectors. Cells were transduced by viral particles and selected using selection markers. For stable fluorescence expression, low-expressing cells were sorted by FACSAria III (BD Biosciences) and used for experiments. HT1080 cells expressing both pHluo_M153R-CD63 and mCherry-CaaX were sorted for red fluorescence by FACSAria III after transducing mCherry-CAAX in HT1080-pHluo_M153R-CD63. Primary antibodies were: anti-CD63 (ab68418, abcam, 1:500 for WB and ab8219, abcam, 1:100 for IF), anti-GFP (A11122, Invitrogen, 1:5000 for WB and 1:500 for immunogold TEM), anti-TSG101 (ab30871, abcam, 1:1000 for WB and 1:100 for IF), anti-flotillin (610820, BD Biosciences, 1:1000 for WB), anti-HSP70 (sc-373867, Santa Cruz Biotechnology, 1:1000 for WB), anti-GM130 (610822, BD Biosciences, 1:250 for WB), anti-Rab27a (69295, Cell Signaling, 1:1000 for WB), anti-Alix (2171, Cell Signaling, 1:200 for IF), anti-LAMP1 (555798, BD Biosciences, 1:500 for IF), anti-Rab7 (9367, Cell Signaling, 1:200 for IF), and anti-β-actin (Ac-74, Sigma, 1:5000 for WB). HRP-conjugated goat anti-mouse IgG (W4021, 1:10,000 for WB) or goat anti-rabbit IgG (W4011, 1:10,000 for WB) were purchased from Promega. Alexa Fluor 546-conjugated donkey anti-mouse IgG (A10036, 1:1000 for IF) or goat anti-rabbit IgG (A11010, 1:1000 for IF) and Alexa Fluor 647-conjugated donkey anti-mouse IgG (A31571, 1:1000 for IF) or goat anti-rabbit IgG (A21244, 1:1000 for IF) were purchased from Invitrogen. Colloidal gold (10 nm)-conjugated donkey anti-rabbit IgG was purchased from Electron Microscopy Sciences (25704, 1:40 for immunogold TEM). Pierce ECL Western Blotting Substrate and Super-Signal WestFemto Maximum Sensitivity Substrate (32106 and 34095, respectively, Thermo Fisher Scientific) were used as substrates for Western blotting. Blots were imaged using an Amersham Imager 600 (GE Healthcare Life Sciences) and analyzed using Fiji.

**Site-directed mutagenesis and cloning**. Methionine153 on pHluorin in pcDNA3.1-pHluorin-CD63[20] was mutated to Arginine using QuickChange II XL Site-Directed Mutagenesis Kit (Agilent) with a pair of primers (Forward, 5′-ACG AGC ACT TGG TGT ACA TCC GGG CAG ACA AAC AAA AGA ATG-3′ and Reverse, 5′-CAT TCT TTT GTT TGT CTG CCC GGA TGT ACA CCA AGT GCT CGT-3′). pcDNA3.1-pHluo_M153R-CD63 was subcloned into pENTR/D-TOPO (Thermo Fisher Scientific) and then cloned into pLenti CMV Blast DEST (706-1) vector (a gift from Eric Campeau and Paul Kaufman, Addgene plasmid #17451) using Gateway recombination cloning (Thermo Fisher Scientific). mCherry-CaaX from pME-mCherry-CaaX (a generous gift from Chi-Bin Chien, University of Utah) was subcloned into pENTR/D-TOPO and then cloned into pLenti6/V5-DEST plasmid using Gateway recombination cloning (Thermo Fisher Scientific). mScarlet from pCytERM-mScarlet_N1 (a gift from Dorus Gadella, Addgene plasmid #85066) was cloned to the C-terminus of pHluo_M153R-CD63 in pLenti CMV Blast DEST (706-1) using Gibson Assembly Master Mix (New England Biolabs Inc.) with two pairs of primers (Forward, 5′-ACG AGC TGT ACA AGG GAT CCT AGA AGG GTG GGC GCG C-3′ and Reverse, 5′-CCC TTG CTC ACC ATG AAT TCC ATC ACC TCG TAG CCA CTT CTG A-3′ for pLenti CMV Blast DEST-pHluo_M153R-CD63 and Forward, 5′-GAA GTG GCT ACG AGG TGA TGG AAT TCA TGG TGA GCA AGG GCG-3′ and Reverse, 5′-TCG GCG CGC CCA CCC TTC TAG GAT CCC TTG TAC AGC TCG T-3′ for pCytERM-mScarlet_N1). Site-directed mutagenesis and all subclonings were confirmed by sequencing (Genewiz).

**Isolation of EVs**. To collect conditioned media, 80% confluent cells were cultured for 48 h in Opti-MEM (Thermo Fisher Scientific). EVs were isolated from conditioned media by serial centrifugation at $300 \times g$ for 10 min, $2000 \times g$ (4000 rpm in Ti45 rotor) for 30 min, $10,000 \times g$ for 30 min (9300 rpm in Ti45), and $100,000 \times g$ (30,000 rpm in Ti45 rotor) overnight to respectively sediment live cells, dead cells, debris and large EVs, and small EVs. Pellets of large and small EVs were resuspended in phosphate-buffered saline (PBS) and spun again in same conditions. Each pellet was resuspended in PBS and used for NTA using ZetaView (Particle Metrix) or for Western blot analysis. At the time of conditioned media collection, cells were trypsinized and counted to allow for estimation of the small EV secretion rate as number of small EVs divided by the number of cells and by the number of hours of media collection (48 h). For further purification of small EVs, the discontinuous gradient of iodixanol (40% (w/v), 20% (w/v), 10% (w/v) and 5% (w/v)) were made by diluting OptiPrep (60% (w/v) aqueous iodixanol, Axis-Shield PoC) with 0.25 M sucrose/10 mM Tris, pH 7.5 from the bottom to the top of a 14 × 89 mm polyallomer tube. Resuspended pellets from $100,000 \times g$ spin were added on a top of the gradient and continuous gradient was made at $100,000 \times g$ (24,000 rpm in SW40 Ti rotor) for 18 h. Twelve fractions were collected and each fraction was diluted in PBS. Each fraction was pelleted through another round of ultracentrifugation at $100,000 \times g$ for 3 h and resuspended in PBS. EV number and size were measured by ZetaView (Paticle Metrix).

**Immunogold transmission electron microscopy**. Formvar-coated grids (Electron Microscopy Sciences) were soaked in deionized water and in absolute EtOH, briefly. Density-gradient-purified small EV suspension in PBS were spotted on the grids for 15 min, fixed with 4% paraformaldehyde for 10 min, and washed with PBS. Free aldehyde groups were quenched with 0.15 M glycine in PBS. The grids were blocked with 5% bovine serum albumin (BSA) in PBS and sequentially incubated with rabbit anti-GFP antibody and colloidal gold-conjugated donkey anti-rabbit IgG. After washing with PBS and deionized water, the grids were stained with 2% phosphotungstic acid, pH 6.1 and allowed to air-dry. Immunogold-labeled and negative-stained small EVs were imaged on a FEI Tecnai T12 Transmission Electron Microscope at 100 kV using an AMT CR41 side mounted CCD camera.

**Immunofluorescence staining**. Cells on coverslips coated with FN (1 µg ml$^{-1}$ overnight at 4° C) were permeabilized with 0.2% Triton X-100 or saponin in PBS after fixation with 4% paraformaldehyde in PBS. After blocking with 5% BSA in PBS, primary and secondary antibodies listed above were sequentially incubated with the cells with PBS washes in between. After mounting coverslips on glass slides, Z-stack images were acquired with an LSM 510 laser scanning confocal microscope (ran by ZEN software, CarlZeiss) equipped with a 63x/1.40 NA Plan Apo oil objective lens or with a Nikon A1R-HD25 confocal microscope (ran by NIS-Elements) equipped with an Apo TIRF 60x/1.49 NA oil immersion lens.

**Correlative light-electron microscopy**. Cells were plated on high Grid-500 glass-bottom µ-Dishes (ibidi) coated with FN (1 µg ml$^{-1}$) and cultured at 37 °C incubator with 5% CO$_2$ for 1 day. Cells were initially washed in 0.1 M sodium cacodylate buffer then briefly fixed in 2% paraformaldehyde. pHluo_M153R-CD63-overexpressing cells were identified using a Nikon Plan Apo 60x/1.40 NA oil immersion lens in a Nikon Eclipse TE2000E microscope equipped with a cooled charge-coupled device (CCD) camera (Hamamatsu ORCA-ER). The cells were then fixed in 2.5% glutaraldehyde in 0.1 M cacodylate buffer, pH7.4 at room temperature (RT) for 1 h then transferred to 4 °C, overnight. The samples were again washed in 0.1 M cacodylate buffer, then incubated 1 h in 1% osmium tetroxide at RT. Subsequently, the samples were dehydrated through a graded ethanol series and three exchanges of 100% ethanol, and then the glass-bottoms were taken off from the dishes, followed by critical point drying using a Tousimis samdri-PVT-3D critical point dryer. The dried coverslips

were then mounted on an aluminum stub with a carbon adhesive tab and sputter coated with gold palladium using the Cresssington Sputter Coater 108. After coating for 60 s, the samples were imaged using a Quanta 250 SEM.

**Live imaging in 2D and 3D.** Cells were plated on glass-bottom MatTek dishes coated with fibronectin (1 µg ml$^{-1}$) and maintained in complete media in a 37 °C incubator with 5% $CO_2$. The next day, the media was changed to Leibovitz's L-15 (Gibco)/10% serum or FluoroBrite DMEM (Gibco)/10% serum depending on the $CO_2$ microscope environment. Live imaging of mCherry-CaaX/pHluo_M153R-CD63-expressing HT1080 cells was performed with a Nikon Eclipse TE2000E epi-fluorescence microscope (ran by MetaMorph software, Molecular Devices) equipped with a 37 °C chamber and a cooled CCD camera using a Nikon Plan Fluor oil 40x/1.30 NA or Super Fluor 20x/0.75 NA objective lens. Still images of pHluo_M153R-CD63 were captured from diverse live cells using a Nikon Plan Fluor oil 40x/1.30 NA lens on the Nikon Eclipse TE2000E (ran by MetaMorph software, Molecular Devices). Time-lapse movies of pHluo_M153R-CD63-expressing cells on nanopatterned glass-bottom dishes (800 nm width of both ridge and groove and 600 nm depth, Nano-surface Biomedical) coated with fibronectin (1 µg ml$^{-1}$), and of cells expressing the dual reporter pHluo_M153R-CD63-mScarlet, were acquired with a Nikon A1R confocal microscope (ran by NIS-Elements) equipped with a Tokai Hit Incubation Chamber (37 °C with 5% $CO_2$) using a Plan Apo 40x/1.3 NA oil immersion lens.

**Intravital imaging.** For mouse intravital imaging, all mouse imaging and surgical protocols were approved by the University of Wisconsin Institutional Animal Care and Use Committee (IACUC) and were carried out in accordance with IACUC guidelines. Complete animal care facilities are provided by the UW Research Animal Resources Center (RARC). RARC personnel provide basic care, and veterinary care is available on site. The facility provides temperature and light-controlled housing for animals, which is operated under the supervision of a full-time veterinarian. For intravital experiments, $1 \times 10^6$ MDA-MB-231 breast cancer cells expressing pHluo_M153R-CD63-mScarlet were injected into the mammary fat pads of 8-week-old female NOD/SCID mice. Imaging of mammary tumors was accomplished through surgical implantation of a mammary imaging window (MIW). Tumors were palpable at 1-week post injection and MIWs were implanted at this time. Multiphoton microscopy of the pHluorin tumors commenced 2 days post MIW implant and was conducted on an Ultima IV (Bruker Nano Surfaces, Middleton, WI) using a Coherent Chameleon laser and Hamamatsu multi-alkali photomultiplier detectors. Data acquisition and scanning control was provided by PrairieView v5.4 (Bruker Nano Surfaces, Middleton, WI). Mice were anesthetized using isoflurane, provided subcutaneous PBS hydration, and placed in a warming chamber mounted to the microscope stage for imaging. Respiration rates were monitored during the imaging period. Images were collected using a Nikon 40x/1.15 NA water immersion objective lens. Imaging of collagen and cells was per-formed simultaneously using a Chroma SHG/eGFP (430/40, 520/40) custom filter cube and 890 nm excitation.

For avian embryo intravital imaging, fertilized chicken embryos were placed in a 37 °C incubator at 55% relative humidity on day 1 post fertilization. Eggs were decanted into sterilized weigh boats on day 3 for ex ovo chorioallantoic membrane development. On day 10, $1 \times 10^5$ HT1080 fibrosarcoma cells expressing pHluo_M153R-CD63/ mCherry-CaaX were injected (100 µl) in the direction of blood flow into the allantoic vein. Twenty-four hours post injection, the ex ovo chicks were transferred to an intravital imaging chamber and placed under a temperature-controlled (37 °C) microscope. Images of HT1080 cells in the chick chorioallantoic membrane were acquired using an Olympus BX61 equipped with Olympus 20x/0.70 NA UApo/340 water immersion objective lens and a CCD camera (Hamamatsu ORCA-ER C4742-80-12AG) controlled with Volocity image acquisition software (PerkinElmer).

**Measurement of in vivo fluorescent half-life of pHluo_M153R-CD63-positive EVs.** In vivo fluorescent half-life of the pHluo_M153R-CD63 marker was deter-mined using the avian embryo model. Fertilized chicken eggs were placed in 37 °C incubator at 55% relative humidity on day 1 post fertilization. Days 12−14 embryos were handled to locate and mark the allantoic vein and the direction of blood flow. A small window was cut into the eggshell using a Dremel drill. pHluo_M153R-CD63 sEV prep was diluted 1:10 in 0.22 µm sterile filtered DPBS (without Ca/Mg) for a final concentration of $6.2 \times 10^7$ EVs µL$^{-1}$. Blood was drawn prior to pHluo_M153R-CD63 small EV injection, designated as time 0 min, into tubes with a final con-centration of 5 mM ethylenediaminetetraacetic acid (EDTA). One hundred micro-liters of diluted pHluo_M153R-CD63 small EV was injected in the direction of blood flow. Blood was then collected in EDTA tubes immediately after injection (1 min) and 30, 60, 120, and 240 min after injection. Plasma was processed upon collection at each time point ($500 \times g$ spin for 10 min) and kept on ice in the dark until use that day. Processed plasma samples and prediluted pHluo_M153R-CD63 small EV (1:50, 1:100, 1:200, and 1:400 in 0.22 µm sterile filtered DPBS [without Ca/Mg]) were diluted 1:2 with a flow cytometry buffer and kept at room temperature in the dark during analysis. Flow cytometry was performed using the CellStream® (Amnis®). All samples were acquired in Small Particle Mode (slow flow rate = 3.66 µL min$^{-1}$ with no thresholds and triggering on all channels). FSC/SSC lasers were set to 0% power and 488 laser power was set to 100%. Stop criteria was set to time for 60 s. Data

analysis and gating was performed in Cytobank by using pHluo_M153R-CD63 small EVs in buffer as a positive control and the preinjection time 0 min plasma as a negative control (Supplementary Fig. 4). Relative percentage of pHluo_M153R intensity at each time points was calculated by comparing with the intensity at 1 min. Half-life of pHluo_M153R-CD63 small EVs was calculated by one phase exponential decay equation in GraphPad Prism.

**Kymograph analysis.** Cells expressing pHluo_M153R-CD63-mScarlet were plated on glass-bottom MatTek dishes coated with fibronectin (1 µg ml$^{-1}$) and time-lapse movies were acquired every 10 s with an A1R-HD25 confocal microscope (ran by NIS-Elements, Nikon) equipped with a Tokai Hit Incubation Chamber (37 °C with 5% $CO_2$) using a Plan Apo 40x/1.3 NA oil immersion lens. Region of interest was selected by a rectangular selection tool and a time-series montage was made by using Fiji (Image/Stacks/Make Montage).

**Image analyses.** To analyze extracellular exosome deposits, cell bodies were carefully selected and deleted from each image. The remaining pHluo_M153R-CD63 or pHluo_M153R-CD63-mScarlet deposits were segmented from the background by thresholding and then measured for area and integrated intensity using Fiji (Analyze tab/Measure). For colocalization of pHluo_M153R-CD63 with costained proteins, fluorescence signals surrounding cells were segmented from the background by thresholding and measured for colocalization of pHluo_M153R-CD63 with TSG101, Alix, or CD63 using MetaMorph (Molecular Devices, Apps/Measure Colocalization) or Fiji (Analyze tab/Colocalization/Colocalization Threshold). For triple colocalization analysis, colocalized points of double colo-calization images were first created using Fiji (Plugins tab/Colocalization) and colocalization with a third staining was measured using Fiji (Analyze tab/Colo-calization/Colocalization Threshold).

To analyze intracellular immunofluorescence staining, staining was segmented from the background by thresholding and then measured for colocalization of pHluo_M153R-CD63-mScarlet with costained proteins using Fiji.

To measure pathfinding Index, cell migration trajectories were created using Fiji (Plugins/Tracking/Manual Tracking). Then, the degree of angle (°) was measured between each trajectory and the closest exosome trail using the Angle Tool in Fiji. Degree was converted into radian and cosine value of radian was calculated using Excel (Microsoft), e.g. cos 0° = 1, cos 90° = 0, and cos 180° = −1. Only single migratory cells were selected for the analysis.

To track and analyze mobility of pHluo_M153R-CD63 deposits on nanopatterned dishes, Imaris (Bitplane, vesicle tracking algorithm) and Fiji (Plugins/Mosaic/Particle Tracker 2D/3D) were used. Immobile deposits were defined as deposits with displacement length < 2 µm during 1 h.

**Numbers and statistics.** For both quantitated data and representative images from experiments, the *n* values and independent experiment numbers are listed in the figure legends. Quantitated imaging data were acquired from at least three independent experiments and multiple images or movies on multiple cells were captured. Cell numbers to be quantitated for each experiment were determined by our experience and data were excluded only if there was an obvious reason for poor data, such as dead or sick-looking cells. For nonquantitated Western blots (e.g. checking knockdown), they were generally performed a single time. All datasets were tested for normality using the Kolmogorov−Smirnov or Shapiro−Wilk normality test in GraphPad Prism. Nonparametric data groups were compared by the two-sided unpaired Mann−Whitney or two-sided paired Wilcoxon test. Parametric data were compared using two-sided unpaired or paired Students' *t* test. Scatter plots were plotted with median and interquartile and bar graphs were plotted as mean ± standard error. The violin plot with median and interquartile range was created by Graphpad Prism to show all data.

**Reporting summary.** Further information on research design is available in the Nature Research Reporting Summary linked to this article.

## Data availability
The data supporting the findings of this study are either available within the supplementary information files and/or the source data file, or are available from the corresponding author upon reasonable request. The source data underlying Figs. 1c, d, 2a–e, g, h, 3b, 4b, d, g, 5b, c, 6b, d, f, g and 7c, d, and Supplementary Figs. 1a, b and 4e are provided as a Source Data file.

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

## Acknowledgements

This study was funded by grants R01GM117916 and R01CA206458 to A.M.W., 3U19CA179514-05S1 to R.J. Coffey, R01CA218526 to A.Z., and R01CA216248 to S.M.P., and supported by vouchers from Vanderbilt CTSA grant UL1 TR002243. Flow cytometry sorting was performed in the Vanderbilt University Medical Center (VUMC) Flow Cytometry Shared Resource. The VUMC Flow Cytometry Shared Resource is supported by the Vanderbilt Ingram Cancer Center (P30 CA68485) and the Vanderbilt Digestive Disease Research Center (DK058404). Data analysis using Imaris, scanning electron microscopy, and transmission electron microscopy was performed in part through the use of the Vanderbilt Cell Imaging Shared Resource (supported by NIH grants CA68485, DK20593, DK58404, DK59637 and EY08126). We thank Ian Macara for use of his Nikon A1R confocal microscope and Matt Tyska, as well as members of the Weaver laboratory for extensive discussion. The content of this article is solely the responsibility of the authors and does not necessarily represent the official views of the National Institutes of Health.

## Author contributions

B.H.S. performed the experiments and data analyses and prepared the figures. A.v.L., J.G., D.I., and R.P. performed experiments. E.S.K. performed EM imaging and aided in experimental design and interpretation. B.H.S., A.Z., S.M.P., and A.M.W designed the experiments. B.H.S. and A.M.W wrote the manuscript, with editing and input from coauthors.

## Competing interests

The authors declare no competing interests.
