## [Peer Review File · Nature Communications]

Reviewers' comments:

Reviewer #1 (Remarks to the Author):

Sung and colleagues expressed pHluorin and mScarlet fused with CD63 in cancer cells to study the secretion and endocytosis of exosomes in real-time. The use of pH-sensitive pHluorin and pH-insensitive mScarlet fluorescent proteins exploits the change in the pH of the MVB lumen during its fusion with the plasma membrane, thereby allowing to track both MVB trafficking and exocytosis. The authors introduced the M153R mutation within pHluorin to enhance the stability and brightness of the signal, thereby increasing resolution compared to its wild type version. While this provides a much improved reporter, the use of this mutation has been described earlier in bacterial systems, so the work is not novel in this regard. While, the microscopy and biochemical analysis of pHluorin_m153R-CD63 positive structures dramatically improved the ability to identify exosomes, the data presented in Figures 1-3 of the ms appear as an extension to the conclusions drawn in a previous paper by the same group (Sung et al, Nature Communications, 2015).

However, the use of mScarlet fusion to CD63-pHluorin does allow the visualization of MVBs within cells along with exosome secretion events, which will be very valuable in the exosome field. While many questions remain unanswered, such as the mechanism of exosome internalization and the fate of internalized exosomes, the ms does provide a novel and powerful tool to study MVB trafficking and exosome turnover.

Minor concerns:

1. How were the exosomes purified? Based on the authors data more than 85% of exosomes are non-motile and form "deposited trails"?
2. Is it the presence of adhesion factors on secreted exosomes or just the preformed tracks on collagen matrix that facilitate the migration of other cells?
3. In Figures 3C, 3E and 3F, why is the pHluorin_M153R-CD63 fluorescence visible within the cells?

Reviewer #2 (Remarks to the Author):

Hwan et al., developed a stable live cell reporter based on a protein associated with small extracellular vesicles (EVs) (e.g. exosomes), to study the secretion and uptake of EVs. This study is a continuation of a previously published study in which the authors developed a live cell reporter based on dim fluorescence. In this study, the authors introduced a mutation in the reporter which conferred stability to the construct, to allow the visualisation of the trafficking of CD63-positive (CD63+) EVs. In addition to the incorporation of a non-pH-sensitive red fluorescent tag to the reporter, the authors also described the exosome (CD63+ vesicles) lifecycle. The extracellular vesicles field is a hot topic in the literature, mainly due to the capacity of these vesicles to facilitate cell-cell communication, via the delivery of bioactive molecules. Thus, this study is novel and innovative, and the development of the "new" live cell reporter which can be used to understand the mechanisms associated with the biogenesis, trafficking and uptake of exosomes, will have significant impact in the field. Interestingly, the authors evaluated the use of this reporter using several approaches, including in-vitro, 3D culture, and in-vivo experiments.

The introduction is well written, and the authors give enough information to the readers to understand the rationale of the study, which leads to the hypothesis and aims of this study. The results are well described, and the conclusions of this study are supported by the results.

In terms of the weakness of this study, I think that the authors should clarify the type of extracellular vesicles that are being targeted with the reporter. As the authors pointed out, CD63 is a protein which has been associated with almost all types of EVs and is not a specific exosomal marker. Thus, the term exosomes should be replaced with CD63+ vesicles, or small vesicles-

CD63+. Furthermore, it has been reported that CD9 is a more specific marker for small vesicles like exosomes. Electron Microscopy of the isolated exosomes needs to be incorporated in to the manuscript.

Can the authors provide information about the reproducibility of the experiments performed? The authors used a crude method to isolate EVs, which has been reported to also contain protein aggregates that can affect the reproducibility of the results.

The other important point is about the content of these vesicles, which was not part of this study, however, could be an interesting avenue to explore. The authors could potentially determine whether transfecting/infecting these cells with the reporter would affect the content of the vesicles. The process of content selection and packaging into vesicles is unclear, however, some RNA binding proteins have been associated with this process. Thus, if the reporter was designed to study the trafficking of exosomes, including the MVB, I am wondering if the changes in the pH might affect the content of exosomes, and thus the molecules involved in this process, which can lead to changes in the trafficking of intraluminal vesicles. Please comment.

In the in vivo experiments, what was the half-life of the vesicles?

Reviewer #3 (Remarks to the Author):

In the paper "pHluo_M153R-CD63, a bright, versatile live cell reporter of exosome secretion and uptake, reveals pathfinding behavior of migrating cells" the authors develop two new sensors a stabilized pHluo_M153R-CD63 for dynamic monitoring of EVs in vivo and a dual color reporter pHluo_M153R-CD63-mScarlet for investigating both MVB secretion and uptake in living cells and organisms. With these two reporters they i) provide the proof of principle that it is possible to track MVB/exosomes lifecycle and trafficking in vivo and ii) identify a role of exosomes in promoting leader-follower behavior in 2D and 3D migration of cancer cells.

There is increasing interest in exosomes not only as new mechanism of cell-cell communication but also for their diagnostic and therapeutic potential. Therefore, these new reporters may be useful tools for investigating the regulation and function of exosomes/MVB secretion and uptake, but also to track them in the organism. From this perspective, the paper is relevant and may be of interest for a broad audience, although the "basic" sensor (pHluo-CD63) is not new, as well as some of the results (ref 20,29-31 of the paper).

If the authors want to propose the new reporters for exosomes studies in vivo, an accurate characterization of the vesicles they mark is necessary.

1) The presence of a tag can interfere with localization and function of the fusion protein. While the pHluo_M153R-CD63 targeting to exosomes has been studied, such a characterization is lacking for the double sensor. This is particularly important because the red protein "mScarlet" was cloned to the CD63 C-terminus which is required for proper sorting and targeting of tetraspanins to specific intracellular domains (Rous et al 2002. doi: 10.1091/mbc.01-08-0409)

2) The reporter is a fusion protein with CD63 which is not only a simple marker but it also directly controls exosomes formation, trafficking and signaling. While exosomes generating from pHluo_M153R-CD63 transfected cells have been characterized in term of dimensions, cargo (double immunofluorescence), genesis and half life, this characterization is lacking for the double sensor.

3) a good sensor should ideally not interfere with cellular processes. Suppl fig1a shows that overexpression of pHluo_M153R-CD63 increases the secretion of exosome-like small EVs. Could this increased secretion affects exosome signaling in living organisms? The authors should

comment this point in the discussion

4) The M153R mutation should increase the recombinant protein stability. However, in Fig.supll1b the anti-GFP staining shows the presence of several degradation products potentially fluorescent (positive for GFP staining). Can they have an impact on the ability of the reporter to label specific mvs subpopulations (Alix vs ESCRT or other vesicles)?

If the authors mean to leverage the novelty of results (mechanisms of exosomes secretion/uptake and role of exosomes in promoting leader-follower behavior in cancer cells migration) they should provide some evidence about the mechanisms by which exosomes act (chemicals released from the exosomes to induce chemotaxis or adhesive/signaling properties of migration tracks)

We thank the reviewers for their time and thoughtful comments. Below is a point-by-point response.

Reviewer #1 (Remarks to the Author):

Sung and colleagues expressed pHluorin and mScarlet fused with CD63 in cancer cells to study the secretion and endocytosis of exosomes in real-time. The use of pH-sensitive pHluorin and pH-insensitive mScarlet fluorescent proteins exploits the change in the pH of the MVB lumen during its fusion with the plasma membrane, thereby allowing to track both MVB trafficking and exocytosis. The authors introduced the M153R mutation within pHluorin to enhance the stability and brightness of the signal, thereby increasing resolution compared to its wild type version. While this provides a much improved reporter, the use of this mutation has been described earlier in bacterial systems, so the work is not novel in this regard. While, the microscopy and biochemical analysis of pHluorin_m153R-CD63 positive structures dramatically improved the ability to identify exosomes, the data presented in Figures 1-3 of the ms appear as an extension to the conclusions drawn in a previous paper by the same group (Sung et al, Nature Communications, 2015).

However, the use of mScarlet fusion to CD63-pHluorin does allow the visualization of MVBs within cells along with exosome secretion events, which will be very valuable in the exosome field. While many questions remain unanswered, such as the mechanism of exosome internalization and the fate of internalized exosomes, the ms does provide a novel and powerful tool to study MVB trafficking and exosome turnover.

Minor concerns:

1. How were the exosomes purified? Based on the authors data more than 85% of exosomes are non-motile and form “deposited trails”?

Based on the request of several reviewers, we have now purified small EVs/exosomes using differential centrifugation+density gradient centrifugation, as described in the Methods section. The new results are in Fig 2 and Supp Fig 5. Our previous results, using just differential centrifugation, are in Fig 1c,d. For the rest of the manuscript, we used imaging of exosome secretion from living cells, so the purification experiments were just for biochemical characterization – no purified EVs were used for imaging experiments. Thus for Fig. 5a-c, we visualized exosomes that are secreted at the basal surface of the cells and for those ~85% were non-motile, meaning that they likely adhered to the substrate surface after secretion. What we cannot image in those experiments are exosomes that are secreted from the apical surface of cells. Nor can we make any statement about what percent of those would be motile or non-motile, as they would float away after secretion. We have added a sentence about this latter point to the Results section on page 10.

2. Is it the presence of adhesion factors on secreted exosomes or just the preformed tracks on collagen matrix that facilitate the migration of other cells?

For the 3D migration experiments, it is of course possible that follower cells are following reorganized collagen tracks/tunnels made by other cells. The collagen was not visualized in our experiments, so we

cannot address whether the exosome trails are concordant with collagen tracks. For the 2D migration experiments, there would not be any collagen tracks, yet we observe a similarly strong pathfinding behavior along exosome trails. Based on our previous work, exosomes contain factors that promote both adhesion and directional movement of cancer cells, both of which could contribute to the pathfinding behavior. We discuss these issues in the fourth paragraph of the Discussion.

3. In Figures 3C, 3E and 3F, why is the pHluorin_M153R-CD63 fluorescence visible within the cells?

We appreciate the opportunity to clarify these points. We do observe some intracellular fluorescent vesicles that likely represent CD63 in early endosomes that have not yet acidified – that this phenomenon occurs is demonstrated in our endocytosis studies with the dual reporter. This may be enhanced in some environments like in 3D gels and in tissues. In addition, there is some out of focus fluorescence likely coming from other cells due to the 3D environment in old Fig 3c (new Fig 4c). Old Fig 3e (new Fig 4e) is a maximum projection of confocal stacks. We have now noted that in the legend and created a 3D reconstruction (new Supp Video 4) from the stack that shows that the majority of pHluorin_M153R-CD63 is localized at the plasma membrane rather than within the cell.

Reviewer #2 (Remarks to the Author):

Hwan et al., developed a stable live cell reporter based on a protein associated with small extracellular vesicles (EVs) (e.g. exosomes), to study the secretion and uptake of EVs. This study is a continuation of a previously published study in which the authors developed a live cell reporter based on dim fluorescence. In this study, the authors introduced a mutation in the reporter which conferred stability to the construct, to allow the visualisation of the trafficking of CD63-positive (CD63+) EVs. In addition to the incorporation of a non-pH-sensitive red fluorescent tag to the reporter, the authors also described the exosome (CD63+ vesicles) lifecycle. The extracellular vesicles field is a hot topic in the literature, mainly due to the capacity of these vesicles to facilitate cell-cell communication, via the delivery of bioactive molecules. Thus, this study is novel and innovative, and the development of the “new” live cell reporter which can be used to understand the mechanisms associated with the biogenesis, trafficking and uptake of exosomes, will have significant impact in the field. Interestingly, the authors evaluated the use of this reporter using several approaches, including in-vitro, 3D culture, and in-vivo experiments.

The introduction is well written, and the authors give enough information to the readers to understand the rationale of the study, which leads to the hypothesis and aims of this study. The results are well described, and the conclusions of this study are supported by the results.

We thank the reviewer for their recognition of our effort to rigorously characterize this valuable new reporter.

In terms of the weakness of this study, I think that the authors should clarify the type of extracellular

vesicles that are being targeted with the reporter. As the authors pointed out, CD63 is a protein which has been associated with almost all types of EVs and is not a specific exosomal marker. **Thus, the term exosomes should be replaced with CD63+ vesicles, or small vesicles-CD63+.** Furthermore, it has been reported that CD9 is a more specific marker for small vesicles like exosomes.

We apologize for not being more clear. We do not think that CD63 is a general marker for EVs, but rather a very good marker for exosomes. In fact, we do not observe it on large microvesicles, obtained classically from a 10,000xg ultracentrifugation (Fig 1d). In addition, it is primarily located in endosomes, with a much smaller amount localized on the plasma membrane, a characteristic to be expected of exosomal proteins. Although we have not ourselves studied CD9, others have shown it to be primarily localized to the plasma membrane, and our general understanding is that it is considered as a possible marker of small microvesicles. Since we have not studied CD9 ourselves, we do not wish to officially comment on that in the manuscript.

We respectfully submit that the term exosome is entirely appropriate and important for the context in which we use it, for the following reasons:

1) The pHluorin approach is specifically designed to report on acidic-to-neutral changes as occurs with MVB fusion with the plasma membrane - and exosomes are the vesicles that come from MVBs. We purposely designed an exosome secretion sensor, not an EV release sensor.

2) CD63 is highly localized to MVB and is a specific marker of exosomes, as demonstrated by the following data:

a) deposition of pHluorin-M153R-CD63+ vesicles from cells is inhibited by Rab27a-KD (Fig 2f-h), which blocks MVB docking with the plasma membrane and exosome secretion (Ostrowski et al., NCB, 2010, Sinha et al., JCB, 2016)

b) pHluorin-M153R-CD63 colocalizes with classic ESCRT MVB machinery-type exosome markers TSG101 and Alix in small EVs (Fig 3a-c)

c) CD63 is used as a classical marker for luminal filling in light and electron microscopy assays of MVB biogenesis (Baietti et al, 2012; Sinha et al., JCB, 2016 among others)

d) a similar pHluorin-CD63 construct was shown by correlative light-transmission electron microscopy to visualize MVB fusion with the plasma membrane (Verweij et al., JCB, 2018).

Electron Microscopy of the isolated exosomes needs to be incorporated into the manuscript.

In response, we have now performed immunogold transmission electron microscopy of the isolated small EVs using an antibody against GFP to demonstrate that the GFP moieties of pHluo_M153R-CD63 and pHluo_M153R-CD63-mScarlet are located at the EV surface as expected (new Fig 2b and Supp Fig 5d). Please note also that these EVs were purified by density gradient centrifugation as per the next comment.

Can the authors provide information about the reproducibility of the experiments performed? The

authors used a crude method to isolate EVs, which has been reported to also contain protein aggregates that can affect the reproducibility of the results.

In response, we have now redone our characterization of the purified vesicles using density gradient fractionation (new Fig 2a-e for pHluo_M153R-CD63 and Supp Fig 5 for pHluo_M153R-CD63-mScarlet. The results are the same.

The other important point is about the content of these vesicles, which was not part of this study, however, could be an interesting avenue to explore. The authors could potentially determine whether transfecting/infecting these cells with the reporter would affect the content of the vesicles. The process of content selection and packaging into vesicles is unclear, however, some RNA binding proteins have been associated with this process. Thus, if the reporter was designed to study the trafficking of exosomes, including the MVB, I am wondering if the changes in the pH might affect the content of exosomes, and thus the molecules involved in this process, which can lead to changes in the trafficking of intraluminal vesicles. Please comment.

We would like to clarify that our reporter does not alter the pH of MVBs, it just has differential fluorescence in acidic and neutral environments based on the pH sensitivity of the fluorophore. It is of course possible that overexpression of CD63 could alter EV cargoes; however we would like to note that the mutation we made should primarily affect the fluorescence as it is in the GFP moiety. Future studies could explore the effect of CD63 overexpression on EV cargo.

To make it more clear how this reporter works, we have added a couple of sentences to the third paragraph of the Introduction. We have also added a sentence about caveats of CD63 overexpression to the second paragraph of the Discussion.

In the in vivo experiments, what was the half-life of the vesicles?

In response, we have performed new experiments to measure the half-life of pHluo_M153R-CD63 EVs in circulation. We used the chick embryo as our in vivo system and performed flow cytometry analysis of the plasma collected at timepoints before and after injection of purified EVs into the chick bloodstream. We find that there is a rapid decay in the number of fluorescent EVs, down to a residual level that is not background, suggesting potential rapid uptake of the majority of EVs. We calculate a half life of 15 min+/-5 min. These data are included in new Fig 4g, with supporting data in Supp Fig 4.

Reviewer #3 (Remarks to the Author):

In the paper “pHluo_M153R-CD63, a bright, versatile live cell reporter of exosome secretion and uptake, reveals pathfinding behavior of migrating cells” the authors develop two new sensors a stabilized pHluo_M153R-CD63 for dynamic monitoring of EVs in vivo and a dual color reporter pHluo_M153R-CD63-mScarlet for investigating both MVB secretion and uptake in living cells and organisms. With these two reporters they i) provide the proof of principle that it is possible to track MVB/exosomes lifecycle

and trafficking in vivo and ii) identify a role of exosomes in promoting leader-follower behavior in 2D and 3D migration of cancer cells.

There is increasing interest in exosomes not only as new mechanism of cell-cell communication but also for their diagnostic and therapeutic potential. Therefore, these new reporters may be useful tools for investigating the regulation and function of exosomes/MVB secretion and uptake, but also to track them in the organism. From this perspective, the paper is relevant and may be of interest for a broad audience, although the “basic” sensor (pHluo-CD63) is not new, as well as some of the results (ref 20,29-31 of the paper).

If the authors want to propose the new reporters for exosomes studies in vivo, an accurate characterization of the vesicles they mark is necessary.

1) The presence of a tag can interfere with localization and function of the fusion protein. While the pHluo_M153R-CD63 targeting to exosomes has been studied, such a characterization is lacking for the double sensor. This is particularly important because the red protein “mScarlet” was cloned to the CD63 C-terminus which is required for proper sorting and targeting of tetraspanins to specific intracellular domains (Rous et al 2002. doi:10.1091/mbc.01-08-0409)

Thank you for your comments. We provide both clarification and additional experiments to address this point.

First, we should note that there are 2 additional amino acids (glutamic acid and phenylalanine) between the C-terminus of CD63 and the mScarlet tag, which should reduce the likelihood of interference with proper sorting of CD63. This information is now noted in the Results and the two amino acids are bolded and underlined in new Supp Fig 5a.

Second, we have now performed a similar characterization for the double labelled sensor as was done for pHluorin-M153R-CD63: NTA, Western blot analysis, and immunogold TEM (new Supp Fig 5).

Third, we performed colocalization of the double sensor with Rab7 and LAMP1 inside the cell (comparing it to either the single tagged sensor or endogenous CD63) (new Fig 6a-d).

Fourth, we performed colocalization of the double sensor (red/green fluorescence) with CD63 in extracellular exosome deposits (new Fig 6e and f).

All of these data suggest that the mScarlet moiety has little effect on CD63 targeting/trafficking.

2) The reporter is a fusion protein with CD63 which is not only a simple marker but it also directly controls exosomes formation, trafficking and signaling. While exosomes generating from pHluorin_M153R-CD63 transfected cells have been characterized in term of dimensions, cargo (double immunofluorescence), genesis and half life, this characterization is lacking for the double sensor.

As mentioned above, we have now done an in depth characterization for the double sensor, shown in new Supp Fig 5, as well as in Fig 6.

3) a good sensor should ideally not interfere with cellular processes. Suppl fig1a shows that overexpression of pHluo_M153R-CD63 increases the secretion of exosome-like small EVs. Could this increased secretion affect exosome signaling in living organisms? The authors should comment on this point in the discussion

We have now commented on this in the Discussion, at the end of the second paragraph.

4) The M153R mutation should increase the recombinant protein stability. However, in Fig.suppl1b the anti-GFP staining shows the presence of several degradation products potentially fluorescent (positive for GFP staining). Can they have an impact on the ability of the reporter to label specific mvs subpopulations (Alix vs ESCRT or other vesicles)?

To address this point, we examined the issue of whether degradation products are present in EVs and affect the reporter as a readout of CD63 secretion. For this purpose, we used both immunoblotting of newly performed density gradient experiments and immunofluorescence, as follows:

1. Further exploration of the possible presence of degradation products in cells versus EVs (new Fig 2c and Supp Fig 5e, compare fraction 6 to the cell lysate lane "L") suggests that the majority of CD63 in EVs is full length and also positive for GFP (major band in both CD63 and GFP blots is ~70 kDa for pHluo_M153R-CD63 EVs and >100 kDa in pHluo_M153R-CD63-mScarlet EVs). In addition, we note that since CD63 is a heavily glycosylated protein and typically runs as a smear on gels, it is difficult to distinguish different glycosylation variants from degradation products. Thus, the lower size of the CD63+/GFP+ bands in cell lysates compared to EVs could represent either less glycosylated proteins (e.g. still in ER) or degradation products (such as might occur due to lysosomal degradation of the construct after fusion of MVBs with lysosomes). This pattern of higher bands for CD63 in EVs compared to cells is common and can be seen in many EV papers that blot for endogenous CD63.

2. In addition to the major band in EVs observed in both the CD63 and GFP immunoblots, there is a prominent band running around 30 kDa that is observed only in the GFP immunoblots and may represent GFP tethered to the EVs through one of the tetraspanin membrane domains (new Fig 2c and Supp Fig 5e, fraction 6). This is indeed likely to represent a degradation product or at least proteolytic clipping of the tetraspanin such that the pieces are separable by SDS-PAGE after lysis of the EVs. Note that a single proteolytic clip could produce such a fragment but not necessarily remove the other parts of CD63 since it is tethered in the membrane by its 4 membrane spanning domains.

3. To explore the role of degradation products in contributing to the extracellular fluorescence signal, we leveraged the dual reporter, pHluo_M153R-CD63-mScarlet, which has pHluorin in the 1st extracellular loop and mScarlet in the C-terminus inside the vesicle – essentially tagging either end of the molecule. We performed colocalization of the respective green and red fluorescence of those two reporters with each other and with immunostaining for CD63. Overall, there was >80% correspondence of any fluorescent signal with the other (new Fig 6f and g), including with CD63. These data indicate that the majority of the extracellular signal marks CD63-positive EVs.

Our conclusion from these experiments is that there is a minor degradation product that produces a GFP-tagged fragment that is slightly larger than the typical 27 kDa GFP MW and may be tethered to the membrane. This fragment does not appear to have much of an effect on its function as a reporter, since even the green-only or red-only signals from the dual reporter are highly colocalized with CD63 and it is similar to colocalization of the dual reporter with CD63. The portion of the CD63+ EVs that do not colocalize with the dual reporter may represent endogenous CD63 expressed by the cell.

If the authors mean to leverage the novelty of results (mechanisms of exosomes secretion/uptake and role of exosomes in promoting leader-follower behavior in cancer cells migration) they should provide some evidence about the mechanisms by which exosomes act (chemicals released from the exosomes to induce chemotaxis or adhesive/signaling properties of migration tracks)

Our goal here was to do a rigorous description of a new tool and show its utility for the field. Manuscripts further elucidating the molecular basis for the pathfinding behavior are an important future direction for us and the topic of future manuscripts.

We thank the reviewers for their helpful comments. We respectfully submit that the manuscript is ready for publication.

REVIEWERS' COMMENTS:

Reviewer #2 (Remarks to the Author):

Thanks to the authors for providing a complete revision of the manuscript. The authors have clarified all my points and performed the modifications accordingly. The current version of the manuscript is clearer and provides enough supporting information for the results described in this study. I do not have more comments.

Reviewer #3 (Remarks to the Author):

The authors develop two "new" live cell reporters (single stable pHluo_M153RCD63 and double pH fluorescent sensor pHluo_M153RCD63-mScarlet) to study the biogenesis, trafficking, uptake and half-life of exosomes.

The new data included in the revised paper convincingly demonstrate that both reporters are correctly targeted to extracellular vesicles and can be used as "bona fide" exosome markers.

The authors also provide the proof of principle that reporters can be used to track exosomes not only in cell culture systems (2D and 3D) but also in vivo.

Given the emerging role of exosomes as new intercellular communication systems and for their diagnostic and therapeutic potential, the new reporters promise to have an important impact on future researches.

Carla Perego